# *Sipa1* deficiency unleashes a host-immune mechanism eradicating chronic myelogenous leukemia-initiating cells

Yan Xu[1,2], Satoshi Ikeda[2], Kentaro Sumida[2], Ryusuke Yamamoto[1,2], Hiroki Tanaka[2] & Nagahiro Minato[1,2]

Chronic myelogenous leukemia (CML) caused by hematopoietic stem cells expressing the *Bcr-Abl* fusion gene may be controlled by Bcr-Abl tyrosine kinase inhibitors (TKIs). However, CML-initiating cells are resistant to TKIs and may persist as minimal residual disease. We demonstrate that mice deficient in *Sipa1*, which encodes Rap1 GTPase-activating protein, rarely develop CML upon transfer of primary hematopoietic progenitor cells (HPCs) expressing *Bcr-Abl*, which cause lethal CML disease in wild-type mice. Resistance requires both T cells and nonhematopoietic cells. *Sipa1*[−/−] mesenchymal stroma cells (MSCs) show enhanced activation and directed migration to *Bcr-Abl*[+] cells in tumor tissue and preferentially produce Cxcl9, which in turn recruits *Sipa1*[−/−] memory T cells that have markedly augmented chemotactic activity. Thus, *Sipa1* deficiency uncovers a host immune mechanism potentially capable of eradicating *Bcr-Abl*[+] HPCs via coordinated interplay between MSCs and immune T cells, which may provide a clue for radical control of human CML.

---

[1] Department of Immunology and Cell Biology, Graduate School of Medicine, Kyoto University, Kyoto 606-8501, Japan. [2] DSK Project, Medical Innovation Center, Graduate School of Medicine, Kyoto University, Kyoto 606-8501 Japan. Correspondence and requests for materials should be addressed to N.M. (email: minato@imm.med.kyoto-u.ac.jp)

Cancer cells may induce dynamic reorganization of their surrounding tissue microenvironment including responses by various host cells, such as mesenchymal stroma cells (MSCs), vascular endothelial cells (ECs), and a variety of immune and inflammatory cells[1–3]. Although such changes in tumor tissue may significantly influence tumor cell growth either positively or negatively depending on the types and extents of tissue reactions, in most cases the tumor cells evolve, expand, invade normal tissues, and eventually metastasize. Acquired immunity has an important role among the changes, and several cellular and molecular signatures of immune contextures in tumor tissue are often associated with a good prognosis for cancer patients[4,5]. Nonetheless, the effects of host immunity, even against potently immunogenic tumors, may be limited due to various hindrances, including immune effectors having ineffective physical and/or functional access to cancer cells in the tissue[6]. The discovery of the PD-1-mediated immune checkpoint mechanism and recent clinical success of checkpoint blockade therapy in cancer patients have highlighted the importance of the functional accessibility of immune T cells to cancer cells in the tissue microenvironment[7–9].

Chronic myelogenous leukemia (CML) is caused by chromosomal translocations resulting in the generation of Bcr-Abl fusion oncogene in a hematopoietic stem cells. Although the Bcr-Abl+ CML-initiating cells proliferate and differentiate under a hematopoietic hierarchy analogous to normal hematopoietic stem/progenitor cells, they overwhelm normal hematopoiesis and lead to a massive increase in the differentiated progenies[10]. The CML cells induce remarkable alterations of the hematopoietic microenvironment, which may favor the survival, proliferation, and differentiation of CML-initiating cells over normal hematopoietic counterparts[11–13]. The robust increase in CML cells during the chronic phase may be controlled by Bcr-Abl tyrosine kinase inhibitors (TKIs), but the CML-initiating cells per se are resistant to TKIs[14,15] and may persist as minimal residual disease[16,17]. It has been reported that T cells specific for CML cells develop in the patients, particularly under certain therapies, which may lead to disease remission[18,19]. However, any contribution of endogenous acquired immunity in the host to the control of CML-initiating cells usually appears to be very limited, if any, allowing the disease progression[20,21]. Therefore, eradication of CML-initiating cells remains a challenge for radical control of CML.

Sipa1 is a specific Rap1 GTPase-activating protein (GAP) that negatively regulates Rap1 signaling[22], which controls cell–cell and cell–matrix interactions through the regulation of cell adhesion-related molecules including integrins[23–25]. Although Sipa1-deficient mice remain healthy for a year or so after birth, they may eventually develop diverse late-onset disorders in the immune and hematopoietic systems, including overt systemic autoimmunity and various hematological disorders with age[26–28]. Such pleiotropic effects of Sipa1 deficiency may be accounted for by the altered cellular interactions involved in the development of various lymphohematopoietic cell lineages in the unique tissue microenvironment[29]. It is reported that certain cancer cells may overexpress Sipa1, which promotes their invasion and metastasis via altered interaction with extracellular matrix (ECM) in the tissue, thus functioning as a metastasis efficiency modifier[30–33]. As such, Sipa1 is a key endogenous regulator of cell adhesion and migration for many cell types in tissues.

To explore the role of the host microenvironment in CML development, we investigated the CML-inducing activity of primary Bcr-Abl+ hematopoietic progenitor cells (HPCs) from the bone marrow (BM) of wild-type (Wt) mice in Sipa1−/− hosts. We demonstrated that Sipa1−/− mice show a remarkable resistance to CML development upon the transfer of Bcr-Abl+ HPCs, which cause lethal CML disease in Wt mice. Current results uncovered a novel mechanism potentially capable of eradicating Bcr-Abl+ HPCs and deterring CML disease development through immune mechanisms via coordinated interplay between MSCs and memory T cells in the tumor tissue.

## Results

**Sipa1-deficient mice resist CML development by Bcr-Abl+ HPCs.** To investigate the effects of the Sipa1-deficient host environment on the CML-inducing activity of the primary Bcr-Abl+ HPCs, we first transferred the primary lineage marker negative (Lin−) BM cells from Wt B6 mice retrovirally transduced with p210Bcr-Abl into Wt and Sipa1−/− B6 mice intravenously at $1.5 \times 10^4$ cells/mouse according to the standard BM transplantation (BMT) procedure[34]. Both Wt and Sipa1−/− mice similarly died within 4 weeks after BMT with lethal or even sublethal (5 Gy) γ-ray irradiation (Fig. 1a, left). Unexpectedly, however, when Bcr-Abl+ HPCs were transferred to unirradiated mice, all Sipa1−/− mice survived with no evidence of disease for more than 100 days (Fig. 1a, right). All Wt mice died within 5 weeks, with only a slight delay compared to irradiated Wt mice (Fig. 1a, right). The Wt mice developed a remarkable increase in GFP+ leukocytes, which mostly consisted of myeloid cells in the blood, and massive splenomegaly, but Sipa1−/− mice showed no detectable GFP+ cells in the circulation and had no splenomegaly (Fig. 1b). To examine the duration of the irradiation effect, we transferred the Bcr-Abl+ HPCs at varying intervals after 5 Gy of irradiation. All Wt recipients died within 4–5 weeks irrespective of the intervals, as expected (Fig. 1c, left). In contrast, 100% and 20% of Sipa1−/− mice that were rested for more than 3 months and for 2 months, respectively, before Bcr-Abl+ HPC transfer survived with no evidence of disease, whereas all the mice with a resting interval of <1 month died within 4 weeks (Fig. 1c, right). The results indicated that the Sipa1−/− host environment contributes potent resistance to the Bcr-Abl+ HPC-mediated CML development, which is radiosensitive and needs at least 3 months for full recovery after irradiation.

**Bcr-Abl+ HPCs are properly engrafted in Sipa-1−/− BM.** We then investigated whether the transferred Bcr-Abl+ HPCs could properly home to hematopoietic tissues and be engrafted in unirradiated Sipa1−/− mice. FACS analysis indicated that a detectable increase in GFP+ cells occurred in the BM of Wt recipients on day 6, followed by an exponential increase thereafter (Fig. 2a). GFP+ cells were then also increased in the peripheral blood (PB) with a delay of ~3 days (Fig. 2a), suggesting that BM was the primary site for the initial expansion and systemic spread of Bcr-Abl+ cells. In Sipa1−/− recipients, GFP+ cells were also increased in the BM until day 9 to an extent comparable to that in Wt mice. However, the increase rate slowed on day 12, and the GFP+ cells became essentially undetectable by day 15; the GFP+ cells in the PB were also transient and disappeared completely by day 15 (Fig. 2a). Immunostaining analysis confirmed the comparable presence of scattered GFP+ cells in the BM parenchyma at as early as day 3, and on day 6 the clustered GFP+ cell foci were observed to be similar in both Wt and Sipa1−/− recipients (Fig. 2b, c). FACS analysis revealed essentially similar multi-lineage differentiation profiles of Bcr-Abl+ HPCs in Wt and Sipa1−/− recipients at day 9 (Supplementary Fig. 1). On day 12, however, there were only minimal residual GFP+ cells in Sipa1−/− BM, whereas remarkable GFP+ cell expansion was evident throughout the BM of Wt mice (Fig. 2b, c). Many of the residual GFP+ cells in Sipa1−/− BM on day 12 were noted to be large cells, suggestive of differentiated megakaryocytes (Fig. 2c). In agreement with these findings, the GFP+ Lin− c-Kit+ Sca-1+ (LKS) progenitor cells that increased until day 9 completely disappeared by day 12 in Sipa1−/− mice, seemingly preceding the

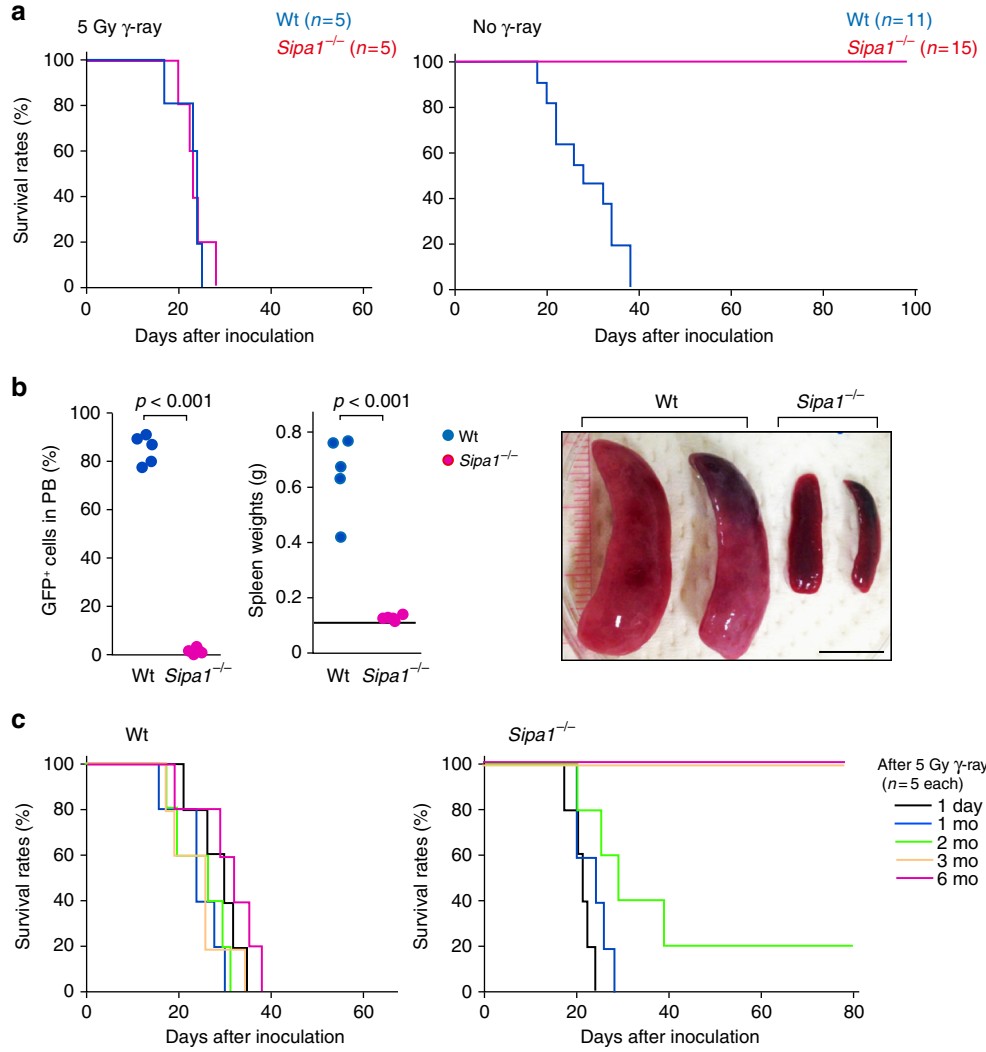

**Fig. 1** $Bcr\text{-}Abl^{+}$ HPCs fail to develop CML in unirradiated $Sipa1^{-/-}$ mice. **a** Normal Lin$^{-}$ BM cells (HPCs) from Wt B6 mice were infected with retrovirus containing $p210Bcr\text{-}Abl$ and intravenously injected into Wt B6 and $Sipa1^{-/-}$ B6 mice at $1.5 \times 10^{4}$ cells per mouse with or without prior γ-ray irradiation. Survival curves of the indicated numbers of mice per group are shown. Independent experiments were performed for two times with similar results. **b** On day 20 after injection of $Bcr\text{-}Abl^{+}$ HPCs, the proportions of GFP$^{+}$ cells in the peripheral blood (PB) leukocytes and the weights of spleens were examined. A bar indicates the mean spleen weight of control mice, and $p$ values were determined with two-tailed unpaired Student's $t$-test. Representative pictures of spleens are also shown. Scale bar, 10 mm. **c** Wt and $Sipa1^{-/-}$ mice were intravenously injected with $Bcr\text{-}Abl^{+}$ HPCs at varying intervals after irradiation with 5 Gy γ-ray. Survival curves of five mice per group are shown. Independent experiments were done twice with similar results

decline of total GFP$^{+}$ cells (Fig. 2d). Further, although BMCs of $Sipa1^{-/-}$ recipients on day 6 were capable of transferring the CML disease to the secondary Wt B6 recipients as with Wt recipients, BMCs of $Sipa1^{-/-}$ recipients on day 18 were no longer able to do so (Fig. 2e). Thus, $Bcr\text{-}Abl^{+}$ HPCs are properly engrafted in the BM and initiated the proliferation and differentiation, but they are eventually rejected completely in the $Sipa1^{-/-}$ host.

**CML resistance requires both hematopoietic and stromal cells.** Next, we addressed the cellular components of the host involved in the rejection of $Bcr\text{-}Abl^{+}$ HPCs in $Sipa1^{-/-}$ mice. Analysis using $EGFP;Sipa1$ reporter mice revealed considerable expression of Sipa1 in the both lymphohematopoietic and nonhematopoietic cells in the BM (Fig. 3a). In the T-cell population, memory CD44$^{high}$ cells exhibited greater Sipa1 expression than naive CD44$^{low}$ cells of both CD4$^{+}$ and CD8$^{+}$ T-cell subsets (Fig. 3a), in agreement with the transcriptional activation of $Sipa1$ via T-cell

receptor (TCR) stimulation[27]. Therefore, we challenged the BM chimeric mice between Wt and $Sipa1^{-/-}$ mice with Wt $Bcr\text{-}Abl^{+}$ HPCs at 4–6 months after BMT, allowing full recovery of the resistance following γ-ray irradiation. As anticipated, <Wt BM into Wt> mice all died within 4 weeks, whereas 100% of <$Sipa1^{-/-}$ BM into $Sipa1^{-/-}$> mice survived with no evidence of disease for more than 100 days (Fig. 3b). All <$Sipa1^{-/-}$ BM into Wt> mice also died similarly to <Wt BM into Wt> mice, indicating that $Sipa1^{-/-}$ hematopoietic cells are ineffective in the Wt host. On the other hand, 3 out of 12 <Wt BM into $Sipa1^{-/-}$> mice (25%) survived for long term, but the rest died similarly to <Wt BM into Wt> mice (Fig. 3b). Although the results may imply that Wt hematopoietic cells contribute to the resistance to a certain extent in the $Sipa1^{-/-}$ host, the effect is much weaker than that of $Sipa1^{-/-}$ hematopoietic cells. With the use of a $Bcr\text{-}Abl^{+}$ CML cell line BA-1, similar results were obtained (Supplementary Fig. 2). The results strongly suggested that the robust resistance of $Sipa1^{-/-}$ mice to CML development depends on both hematopoietic and nonhematopoietic cell components of the

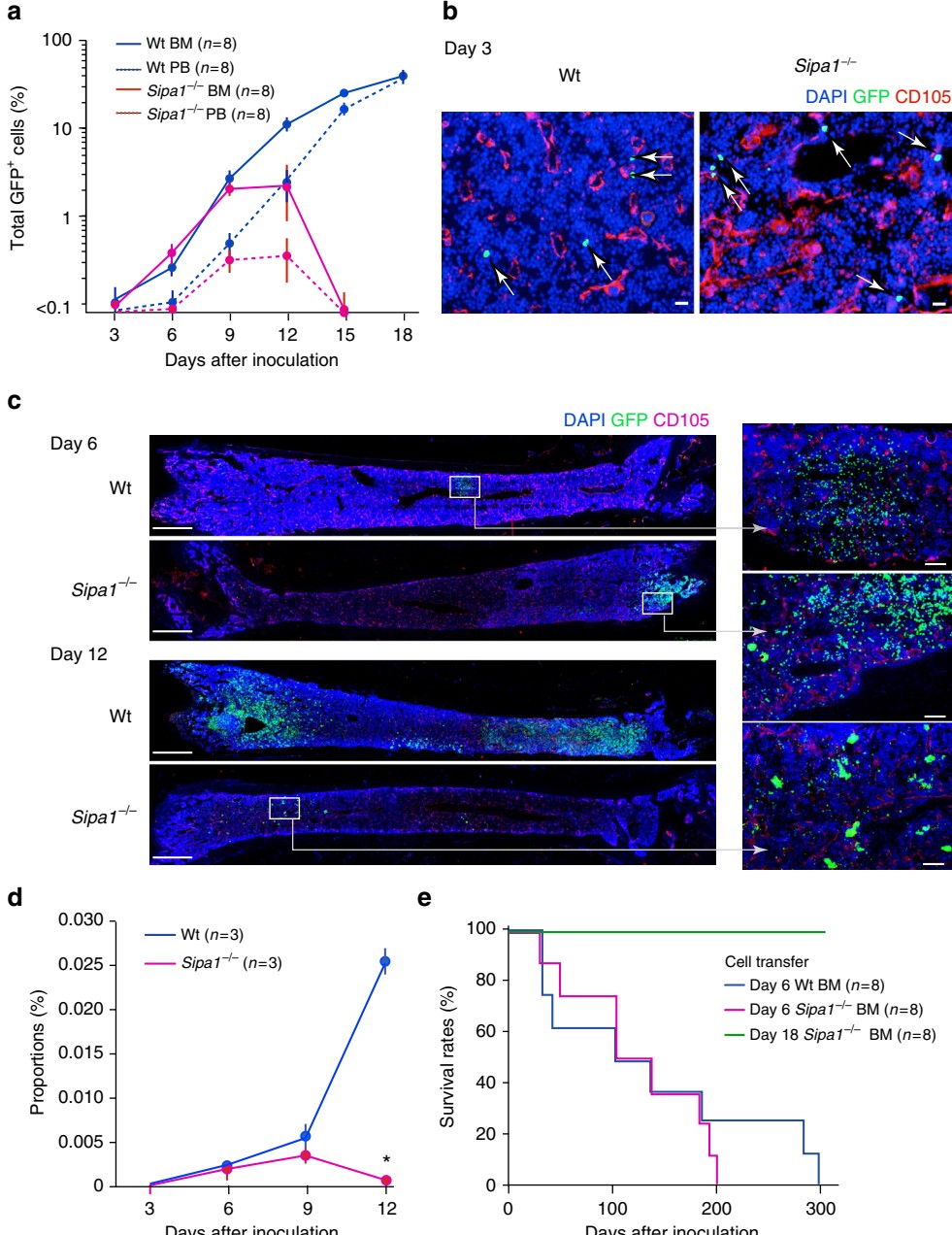

**Fig. 2** *Bcr-Abl*+ HPCs home to and initiate proliferation in BM but are eventually rejected in the *Sipa1*−/− host. **a** *Bcr-Abl*+ HPCs were intravenously injected into unirradiated Wt and *Sipa1*−/− mice, and the proportions of GFP+ cells in the BM and PB were assessed with FACS on varying days after injection. The means and standard errors (SEs) of eight mice are shown. The experiments were performed twice with similar results. **b** On day 3 after injection with *Bcr-Abl*+ HPCs, BMs were immunostained with DAPI, anti-GFP antibody, and anti-CD105 antibody for endothelial cells. Arrows indicate the injected GFP+ cells in the BM parenchyme. Scale bars, 20 μm. Similar results were confirmed in three mice of each group. **c** On days 6 and 12, BMs were immunostained with DAPI, anti-GFP, and anti-CD105 antibodies. The enlarged images of boxed regions are also shown. Scale bars, 1 mm and 100 μm (enlarged). Similar results were obtained in three mice of each group. **d** The proportions of GFP+ Lin− cKit+ Sca-1+ (LKS) cells in the BM of Wt and *Sipa1*−/− mice were assessed with FACS at varying days after injection of *Bcr-Abl*+ HPCs. The means and SEs of three mice per group are shown. *$p < 0.01$ (two-tailed unpaired Student's *t*-test). **e** BM cells from Wt and *Sipa1*−/− mice that received *Bcr-Abl*+ HPCs were harvested on day 6 and day 18, and intravenously injected into the secondary unirradiated Wt B6 mice on a one-to-one basis. Survival curves of eight recipients per group are shown

host. *Sipa1*−/− mice were no more resistant than Wt mice against unrelated leukemia cell lines, such as the T-ALL cell line Wo1, which also expresses GFP, and the T-cell leukemia cell line EL4 (Fig. 3c), and thus the resistance was apparently selective for *Bcr-Abl*+ cells.

**_Sipa1_−/− T cells are essential for the CML resistance.** To identify the hematopoietic cells involved in the resistance of

*Sipa1*−/− mice to *Bcr-Abl*+ HPCs, we performed genetic studies. The resistance of *Sipa1*−/− mice was completely abolished by the introduction of the *Rag2*−/− genetic background (Fig. 4a). *Sipa1*−/− CD3ε−/− mice also developed lethal CML comparably to Wt mice, whereas all *Sipa1*−/−μMT−/− mice remained disease free for more than 100 days (Fig. 4b, c). Notably, both *Sipa1*+/+ *Rag2*−/− and *Sipa1*+/+ CD3ε−/− mice were by no means more susceptible than Wt mice, suggesting that T cells play little role in

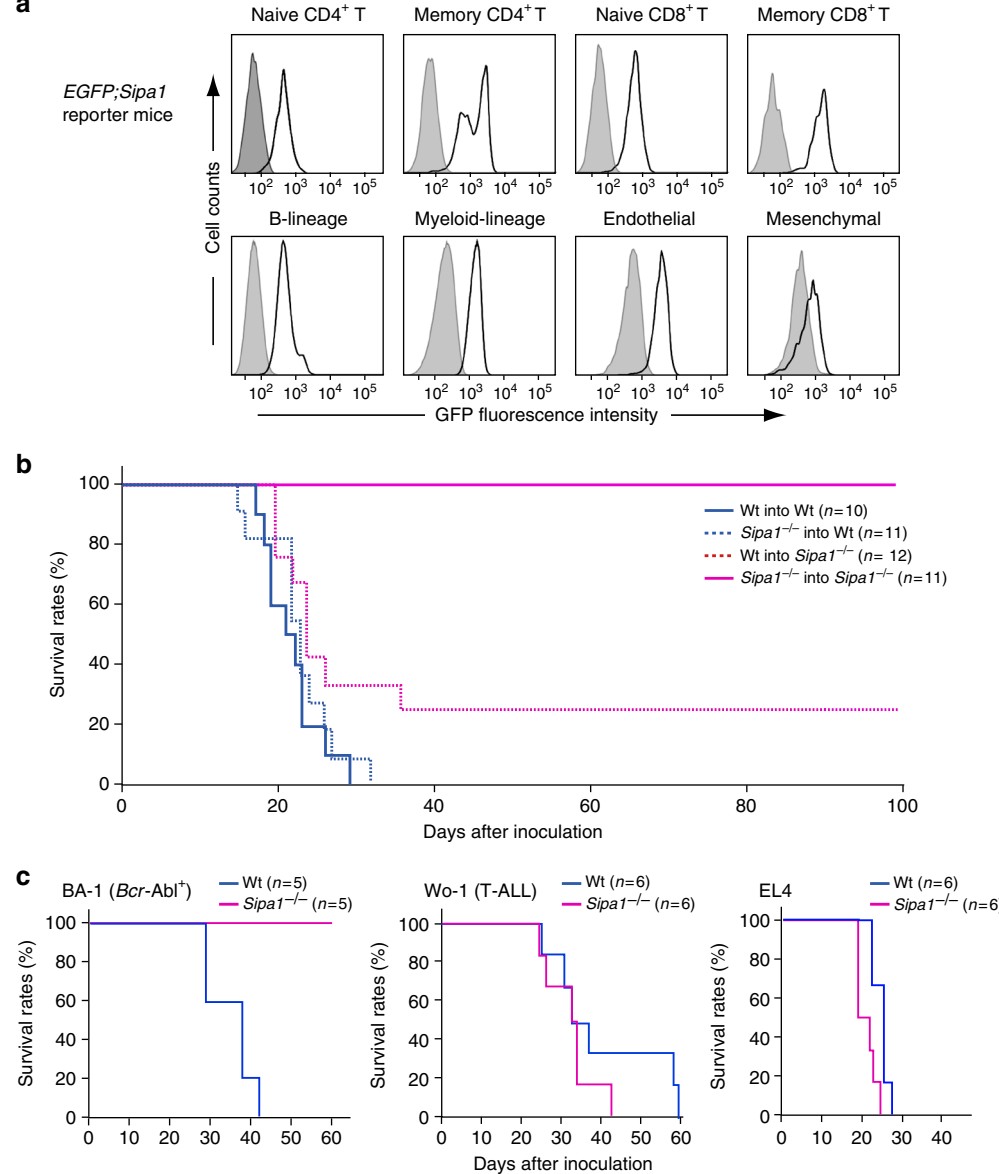

**Fig. 3** CML resistance of *Sipa1*$^{-/-}$ mice requires both hematopoietic and non-hematopoietic cells. **a** Expression of GFP in the BM cells of *EGFP; Sipa1* reporter mice was analyzed with FACS at the gates of CD3$^+$ CD44$^{low}$ CD62L$^{high}$ CD4$^+$ (naive CD4 T), CD3$^+$ CD44$^{high}$ CD62L$^{low}$ CD4$^+$ (memory CD4 T), CD3$^+$ CD44$^{low}$ CD62L$^{high}$ CD8$^+$ (naive CD8 T), CD3$^+$ CD44$^{high}$ CD62L$^{low}$ CD8$^+$ (memory CD8 T), CD45$^+$ B220$^+$ (B-lineage), CD45$^+$ CD11b$^+$ (Myeloid), CD45$^-$ Ter119$^-$ CD31$^+$ (Endothelial), and CD45$^-$ Ter119$^-$ CD31$^-$ PDGFRα$^+$ (Mesenchymal). Shaded regions indicate staining with isotype-matched control IgG. The intensities of GFP were confirmed to correlate with the intracellular Sipa1 expression levels. **b** BM chimeras between Wt and *Sipa1*$^{-/-}$ mice were generated for all four donor/recipient combinations as indicated. At the intervals of 4–6 months after BMT, *Bcr-Abl*$^+$ HPCs of Wt mice were intravenously injected into the chimeric mice at 1.5× 10$^4$ cells per mouse. Survival curves of indicated numbers of recipients per group are shown. **c** Leukemia cell lines of B6 mice, BA-1 (*Bcr-Abl*$^+$ CML), Wo-1 (T-ALL), and EL4 (T cell), were intravenously injected at 10$^5$ cells per mouse into Wt and *Sipa1*$^{-/-}$ mice, and the survival rates were examined. Independent experiments were repeated for at least three times with similar results

the Wt host. To investigate the role of T-cell subsets, we examined the effects of antibody-mediated cell depletion. Depletion of either CD4$^+$ or CD8$^+$ T cells completely abolished the resistance of *Sipa1*$^{-/-}$ mice, although natural killer (NK)-cell depletion had no effect (Fig. 4d). These results indicated that acquired T-cell immunity including both CD4$^+$ and CD8$^+$ T cells has a crucial role in mediating the rejection of *Bcr-Abl*$^+$ HPCs in *Sipa1*$^{-/-}$ mice. In agreement with the findings, *Sipa1*$^{-/-}$ BM at day12 after *Bcr-Abl*$^+$ HPC transfer contained significantly more T cells than Wt BM with less GFP$^+$ cells ($p < 0.01$, two-tailed unpaired Student's *t*-test), the increase being more prominent in CD8$^+$ T cells than CD4$^+$ T cells (Fig. 4e). Also, the majority of increased T cells

were memory cell type (Fig. 4e), suggestive of systemic influx. Immunostaining analysis further confirmed that the foci of GFP$^+$ cells in the *Sipa1*$^{-/-}$ BM were associated with substantially more T cells than those in Wt BM, and such T cells often formed a tight adhesion to GFP$^+$ cells (Supplementary Fig. 3).

**CML resistance of *Sipa1*$^{-/-}$ host is not confined to the BM.** We next addressed the nature of nonhematopoietic cells of *Sipa1*$^{-/-}$ mice in the context of resistance to *Bcr-Abl*$^+$ HPCs. To investigate whether the resistance involves unique hematopoietic stroma cells in *Sipa1*$^{-/-}$ BM, we subcutaneously inoculated *Bcr-Abl*$^+$

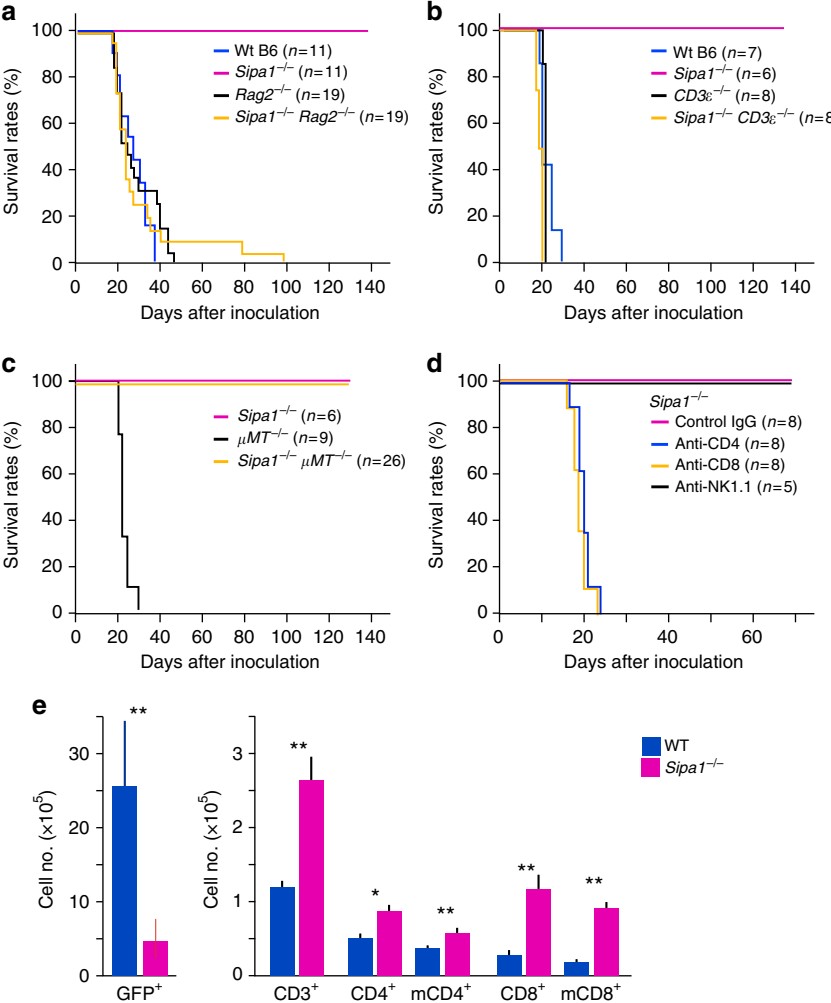

**Fig. 4** T cells of both CD4+ and CD8+ cell types are essential for CML resistance of *Sipa1−/−* mice. **a–c** *Sipa1−/−* mice were crossed with *Rag2−/−*, *CD3ε−/−*, or *μMT−/−* mice to generate double KO mice, and *Bcr-Abl+* HPCs of Wt B6 mice were intravenously injected in the unirradiated mice. Survival curves of the indicated numbers of mice per group are shown. **d** *Sipa1−/−* mice were injected with anti-CD4, anti-CD8, anti-NK1.1 antibody, or control IgG twice per week beginning from the day before the injection of *Bcr-Abl+* HPCs. The antibody treatment was confirmed to eliminate more than 95% of the corresponding cell types in the spleen. Survival curves of the indicated numbers of mice per group are shown. **e** Wt and *Sipa1−/−* B6 mice were intravenously injected with *Bcr-Abl+* HPCs at $1.5 \times 10^4$ cells per mouse, and 12 days later the BM cells were harvested and multi-color analyzed with indicated markers using FACS Canto. The means and SEs of cell numbers of indicated cell populations of five mice per group are shown. *$p < 0.05$; **$p < 0.01$ (two-tailed unpaired Student's *t*-test); m memory phenotype. The similar experiments were repeated two times with similar results

HPCs in Matrigel matrix to prevent systemic cell diffusion. The "ectopic" transfer of *Bcr-Abl+* HPCs into Wt mice also resulted in the local expansion and tumor formation and eventually killed the majority of recipients (Fig. 5a). Significant GFP+ cells appeared in the regional lymph nodes and circulation only at a later stage. The *Bcr-Abl+* HPCs in subcutaneous tissue showed differentiation biased to B-lineage cells unlike in BM, where they were mostly differentiated into myeloid cells (Supplementary Fig. 4). This finding probably reflects the different hematopoietic cytokines available in particular tissues. In contrast, the *Bcr-Abl+* HPCs inoculated subcutaneously into *Sipa1−/−* mice initially formed small nodules but were completely rejected after 3 weeks; 100% of the recipients remained tumor-free for more than 120 days, with no appearance of GFP+ cells in the circulation either (Fig. 5a). Essentially similar results were obtained when *Sipa1−/−* mice were subcutaneously challenged with BA-1 cells, whereas EL4 leukemia cells formed progressive tumors comparably in both *Sipa1−/−* and Wt mice (Fig. 5b). The resistance of *Sipa1−/−* mice to subcutaneous *Bcr-Abl+* HPCs was also

radiosensitive and dependent on T cells, including CD8+ T cells (Supplementary Fig. 5, Fig. 5c). To further investigate the immunological memory, we subcutaneously injected *Sipa1−/−* mice with Matrigel matrix with or without *Bcr-Abl+* HPCs on one frank, followed by a challenge with *Bcr-Abl+* HPCs on the other frank at day 30, when the primary tumors were completely rejected. Although control *Sipa1−/−* mice showed transient tumor growth, those that rejected the tumors developed only negligible tumors, suggesting an effective memory response (Fig. 5d). These results suggested that nonhematopoietic cells required for the rejection of *Bcr-Abl+* HPCs in *Sipa1−/−* mice are not confined to the hematopoietic tissues.

**Infiltration of MSCs and T cells inside CML of *Sipa1−/−* host.** We next performed histological analysis of the subcutaneous tumors. The subcutaneously injected *Bcr-Abl+* HPCs formed tightly packed blastic tumor tissues in Wt mice with some small mononuclear cells at the peripheral edges (Fig. 6a, left). In

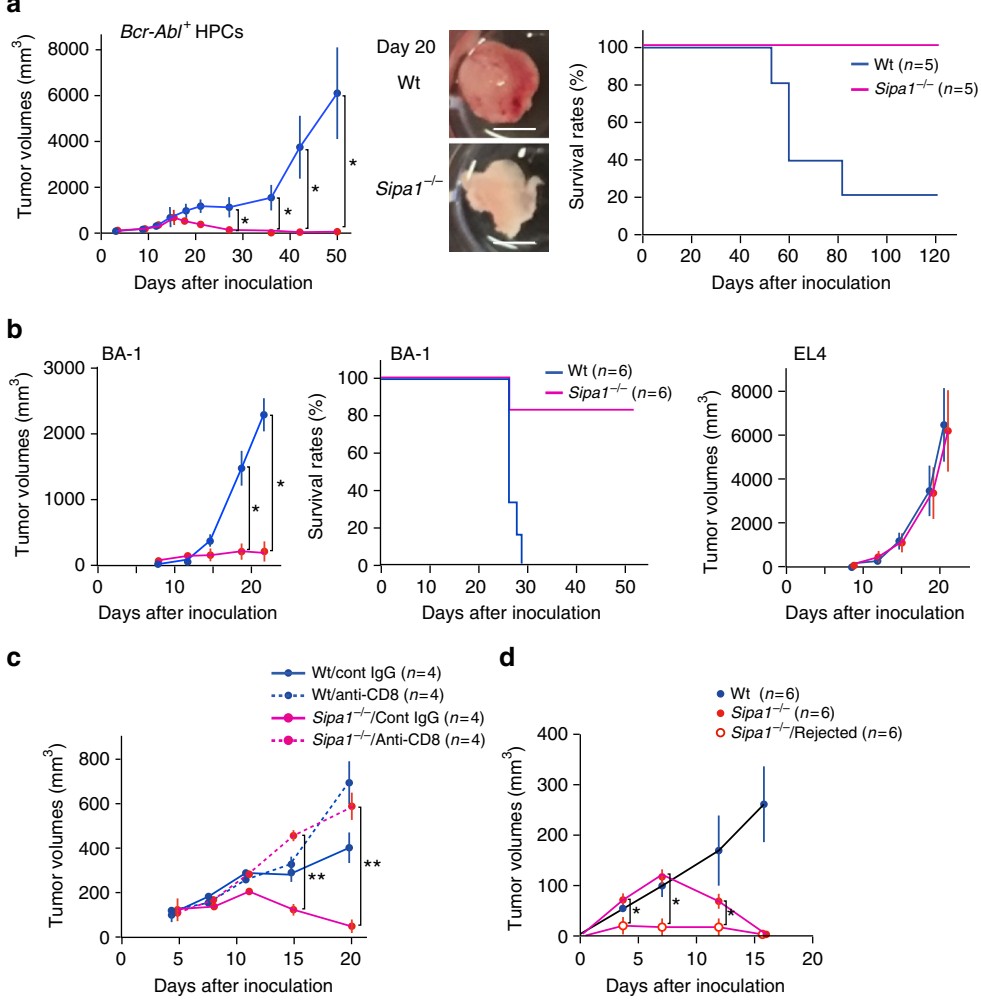

**Fig. 5** $Sipa1^{-/-}$ mice reject the subcutaneous tumors by $Bcr-Abl^+$ HPCs in a T-cell-dependent manner and develop immune memory. **a** $Bcr-Abl^+$ HPCs in Matrigel matrix were injected subcutaneously into Wt and $Sipa1^{-/-}$ mice at $1.5 \times 10^4$ cells per mouse, and the tumor volumes in the local sites were measured. The means and SEs of tumor volumes and survival curves of five mice per group are shown. *$p < 0.05$ (two-tailed unpaired Student's $t$-test). The experiments were done at least three times with similar results. Representative pictures of tumor masses on day 20 are also indicated. Scale bars, 5 mm. The white mass in $Sipa1^{-/-}$ mice represents residual Matrigel matrix. **b** BA-1 or EL4 leukemia cells were subcutaneously injected into Wt and $Sipa1^{-/-}$ mice, and the tumor volumes were measured. The means and SEs of tumor volumes and survival curves of six mice per group are shown. *$p < 0.05$ (two-tailed unpaired Student's $t$-test). The experiments were performed four times. **c** Wt and $Sipa1^{-/-}$ mice were injected with anti-CD8 antibody or isotype-matched IgG, followed by subcutaneous challenge with $Bcr-Abl^+$ HPCs. The means and SEs of tumor volumes of four mice per group are shown. **$p < 0.01$ (two-tailed unpaired Student's $t$-test). The experiments were done twice. **d** $Sipa1^{-/-}$ mice were subcutaneously injected with $Bcr-Abl^+$ HPCs on one side of frank. Thirty days later, when the tumors were completely rejected, these mice were rechallenged with $Bcr-Abl^+$ HPCs on the other side of frank. Untreated Wt and $Sipa1^{-/-}$ mice served as unimmunized controls. The means and SEs of tumor volumes of six mice per group are shown. *$p < 0.05$ (two-tailed unpaired Student's $t$-test)

contrast, tumor tissue in $Sipa1^{-/-}$ mice showed much dispersed tumor cells that was heavily infiltrated with fibroblastic and mononuclear cells inside (Fig. 6a, right). Immunostaining analysis revealed massive accumulation and invasion of vimentin-positive MSCs and CD3$^+$ T cells at largely coinciding areas in the tumor tissues of $Sipa1^{-/-}$ mice, whereas these host cells were scarce in the tumor tissues of Wt mice (Fig. 6b). At a higher magnification, the vimentin-positive cells in $Sipa1^{-/-}$ mice showed spread reticular forms and infiltrating CD3$^+$ T cells were irregularly shaped and often had tails, suggestive of activation (Fig. 6b). FACS analysis confirmed a remarkable increase in the proportions of T cells relative to GFP$^+$ tumor cells in the tissues of $Sipa1^{-/-}$ mice and, to a lesser extent, B cells, macrophages, and dendritic cells, with negligible γδT cells, NK cells, and granulocytes (Fig. 6c). The actual numbers of both CD4$^+$ and CD8$^+$ T cells were also greater in the tumor tissues of $Sipa1^{-/-}$ mice,

the majority showed memory phenotype (Fig. 6d). The ratios of CD8$^+$ to CD4$^+$ T cells tended to be greater in the tumors of $Sipa1^{-/-}$ mice than in those of Wt mice (0.70 vs. 0.35), whereas the proportions of Foxp3$^+$ cells were comparable (about 10%). The results suggested that MSCs play an important role in rejecting $Bcr-Abl^+$ CML cells along with T cells in the tumor tissue of $Sipa1^{-/-}$ mice.

**Activation of $Sipa1^{-/-}$ MSCs and T-cell chemokine secretion.** To assess the activation status of MSCs in tumor tissues, we sorted the GFP$^-$ CD45$^-$ CD31$^-$ Ter119$^-$ cell populations mostly representing mesenchymal cells from $Bcr-Abl^+$ tumors of Wt and $Sipa1^{-/-}$ mice for comparative DNA microarray. Gene set enrichment analysis indicated a marked increase in several gene sets in a $Sipa1^{-/-}$ population, including mesenchymal lineage

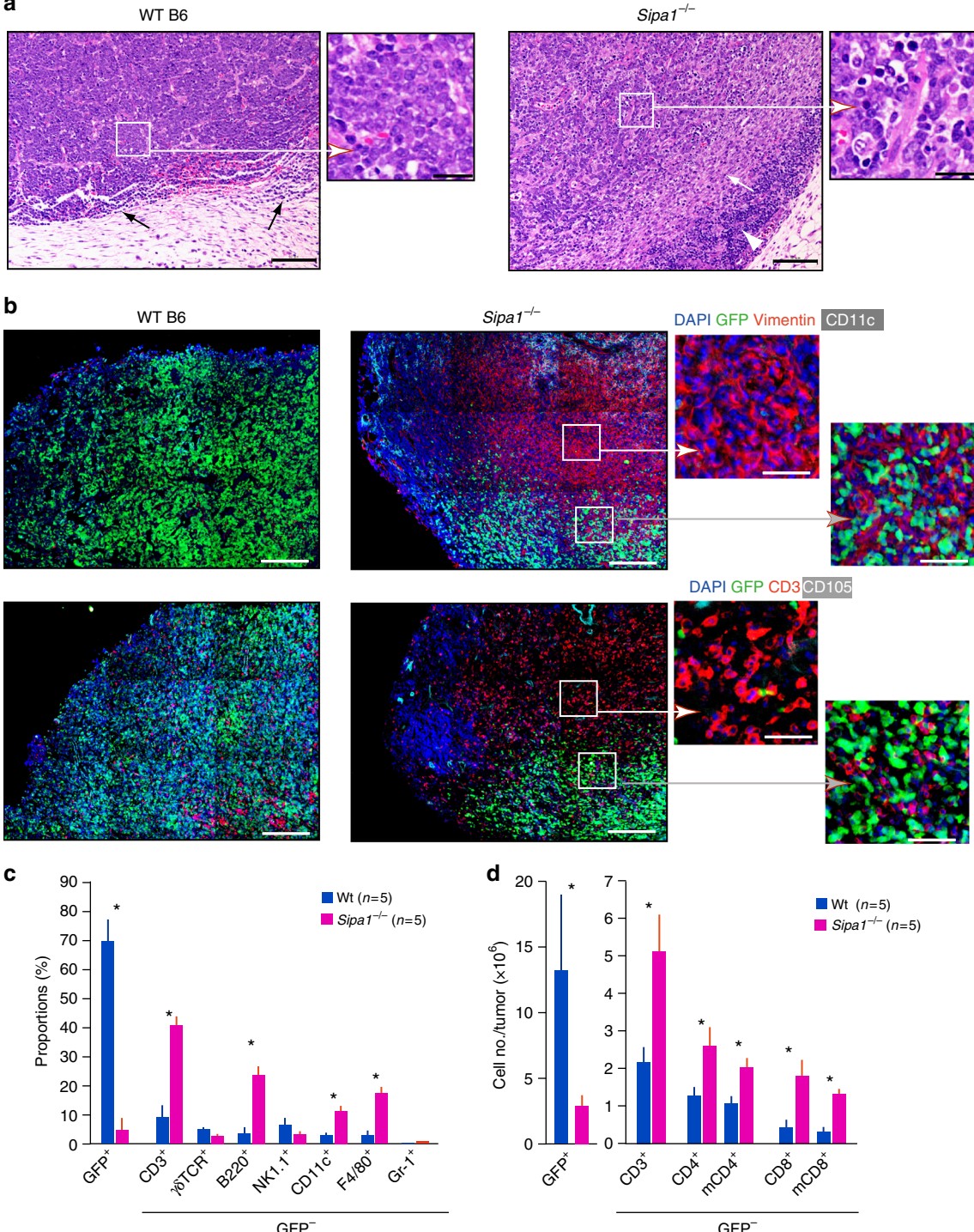

**Fig. 6** Marked increase in memory T-cell infiltration inside *Bcr-Abl*[+] tumors coinciding with increase vimentin[high] MSCs in *Sipa1*[−/−] mice. **a** *Bcr-Abl*[+] HPCs were subcutaneously injected in Matrigel matrix in Wt and *Sipa1*[−/−] mice, and the tumors on day 15 were fixed and stained with hematoxylin and eosin. Black arrows for Wt mice indicate mononuclear cell infiltration in the peripheral edges of the tumor mass. Scale bars, 100 and 20 μm (enlarged). The white arrow and arrowhead indicate stromal reaction and dense mononuclear cell infiltration, respectively. Enlarged images of boxed regions are also shown. **b** Serial sections of subcutaneous *Bcr-Abl*[+] tumors in Wt and *Sipa1*[−/−] mice were immunostained with indicated antibodies. Enlarged images of boxed regions are also shown. Scale bars, 200 and 50 μm (enlarged). **c** Subcutaneous *Bcr-Abl*[+] tumors on day 15 in Wt and *Sipa1*[−/−] mice were dispersed into single-cell suspensions in collagenase/DNase I solution and analyzed for the indicated cell markers with FACS. The means and SEs of the proportions of five mice per group are shown. *$p < 0.01$ (two-tailed unpaired Student's *t*-test). Experiments were done three times with similar results. **d** Subcutaneous *Bcr-Abl*[+] tumors on day 12 in Wt and *Sipa1*[−/−] mice were FACS analyzed, and the actual cell numbers of indicated cell types in tumor tissues were calculated. The means and SEs of five mice per group are shown. m memory phenotype; *$p < 0.01$ (two-tailed unpaired Student's *t*-test). Experiments were done twice with similar results

genes (Supplementary Fig. 6). Using quantitative polymerase chain reaction (qPCR), we confirmed that intratumor $Sipa1^{-/-}$ MSCs at day 7 showed increased expression of mesenchymal lineage genes (*Vimentin, Sca-1, Icam1, Pdgfra, Pdgfrb, Snai1/2*), ECM- and ECM receptor-related genes (*Col1a/3a, Intb1, Intb5, Inta5, Inta11, IntaV*), and fibroblast growth factor (FGF) and FGF receptor (FGFR) family genes (*Fgfr1, Fgf1/2/7/10*) (Fig. 7a). Expression of *α-Sma* characteristic for reactive stroma was also increased but only slightly. Using further purified PDGFRα⁺ MSC populations, essentially similar results were obtained. Activation of $Sipa1^{-/-}$ MSCs was only transient and contracted as the tumor cell burdens began to decrease at a later stage (Supplementary Fig. 7a). In addition, intratumor $Sipa1^{-/-}$ MSCs showed remarkably increased expression of *Cxcl9, Cxcl10*, and *Cxcl12* potentially capable of targeting activated T cells, with minimal expression of other chemokines genes targeting inflammatory myeloid cells (Fig. 7b). To examine actual chemokine secretion in the tumor tissue, we also performed protein array analysis in the tumor tissue fluids. The tumor tissue of

$Sipa1^{-/-}$ mice contained remarkably abundant Cxcl9 compared to those of Wt mice (Fig. 7c). Notably, $Sipa1^{-/-}$ tumor tissue also contained a high level of Ccl5, another potential T-cell chemokine, although the *Ccl5* expression was negligible in $Sipa1^{-/-}$ MSCs (Fig. 7b, c). Intracellular FACS analysis revealed that Ccl5 was produced nearly exclusively by memory CD8⁺ T cells in the tumor tissue of $Sipa1^{-/-}$ mice (Supplementary Fig. 7b). These results suggest that $Sipa1^{-/-}$ MSCs are markedly activated and secrete abundant T-cell chemokines in *Bcr-Abl*⁺ tumor tissue.

**$Sipa1^{-/-}$ MSCs and T cells show enhanced migration activity.** Finally, we investigated the interaction mode of $Sipa1^{-/-}$ MSCs with *Bcr-Abl*⁺ cells using primary mouse embryonic fibroblasts (MEFs) in vitro. $Sipa1^{-/-}$ MEFs showed markedly increased phosphorylation of focal adhesion kinase (FAK), an essential molecular component required for cell migration[35], on collagen I matrix compared to Wt MEFs with unchanged FAK protein level, suggesting enhanced basal ECM-mediated signaling (Fig. 8a,

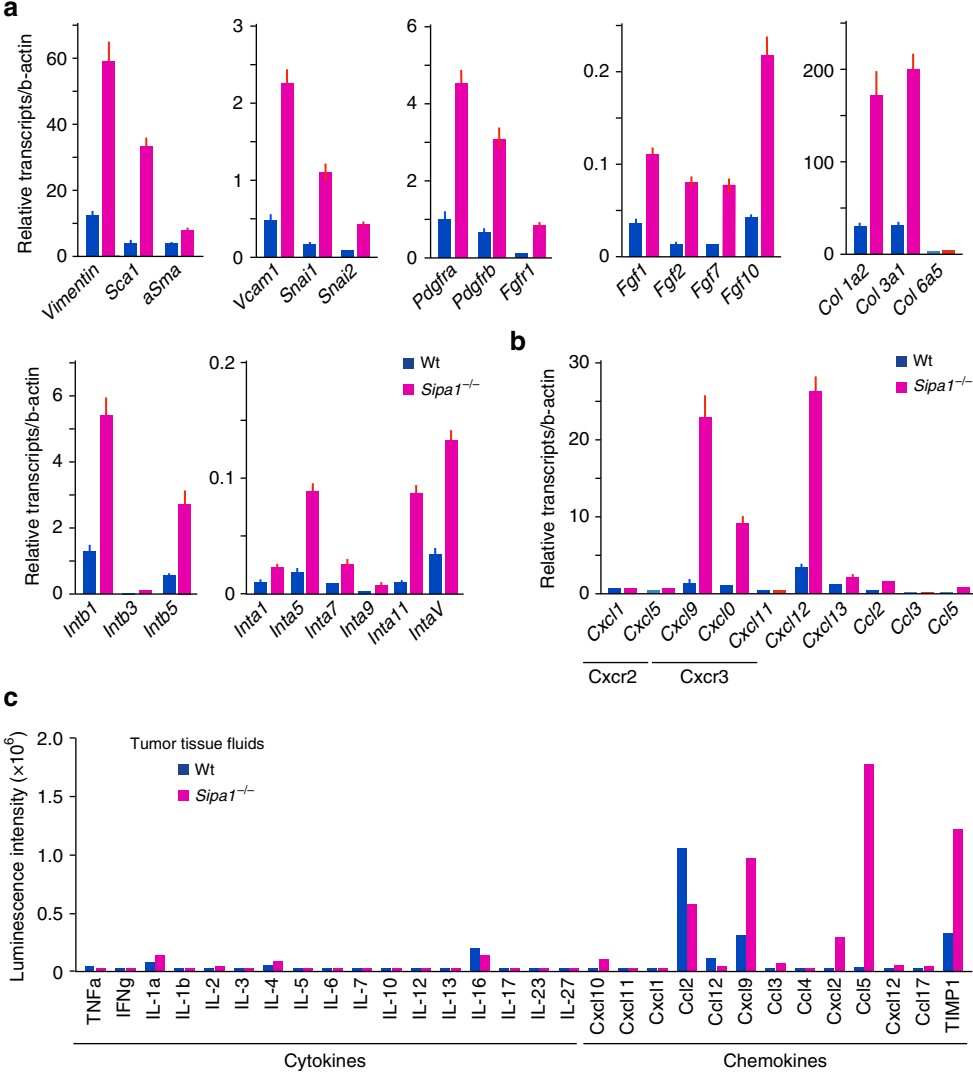

**Fig. 7** $Sipa1^{-/-}$ MSCs in *Bcr-Abl*⁺ tumor tissue show mesenchymal gene activation with preferential production of T-cell chemokines. **a**, **b** MSCs (GFP⁻ CD45⁻ Ter119⁻ CD31⁻) were sorted from the subcutaneous tumors of Wt and $Sipa1^{-/-}$ mice on day 7 after subcutaneous injection of *Bcr-Abl*⁺ HPCs, and the expression of indicated genes was assessed with quantitative PCR. Reanalysis of the sorted cell population indicated that the contamination of GFP⁺ cells was <0.5%. Three to four mice of each group were pooled for an experiment, and independent experiments were done twice with similar results. **c** Cytokines and chemokines in the tissue fluids of *Bcr-Abl*⁺ tumors of Wt and $Sipa1^{-/-}$ mice on day 15 after subcutaneous injection of *Bcr-Abl*⁺ HPCs were assessed using mouse cytokine antibody array

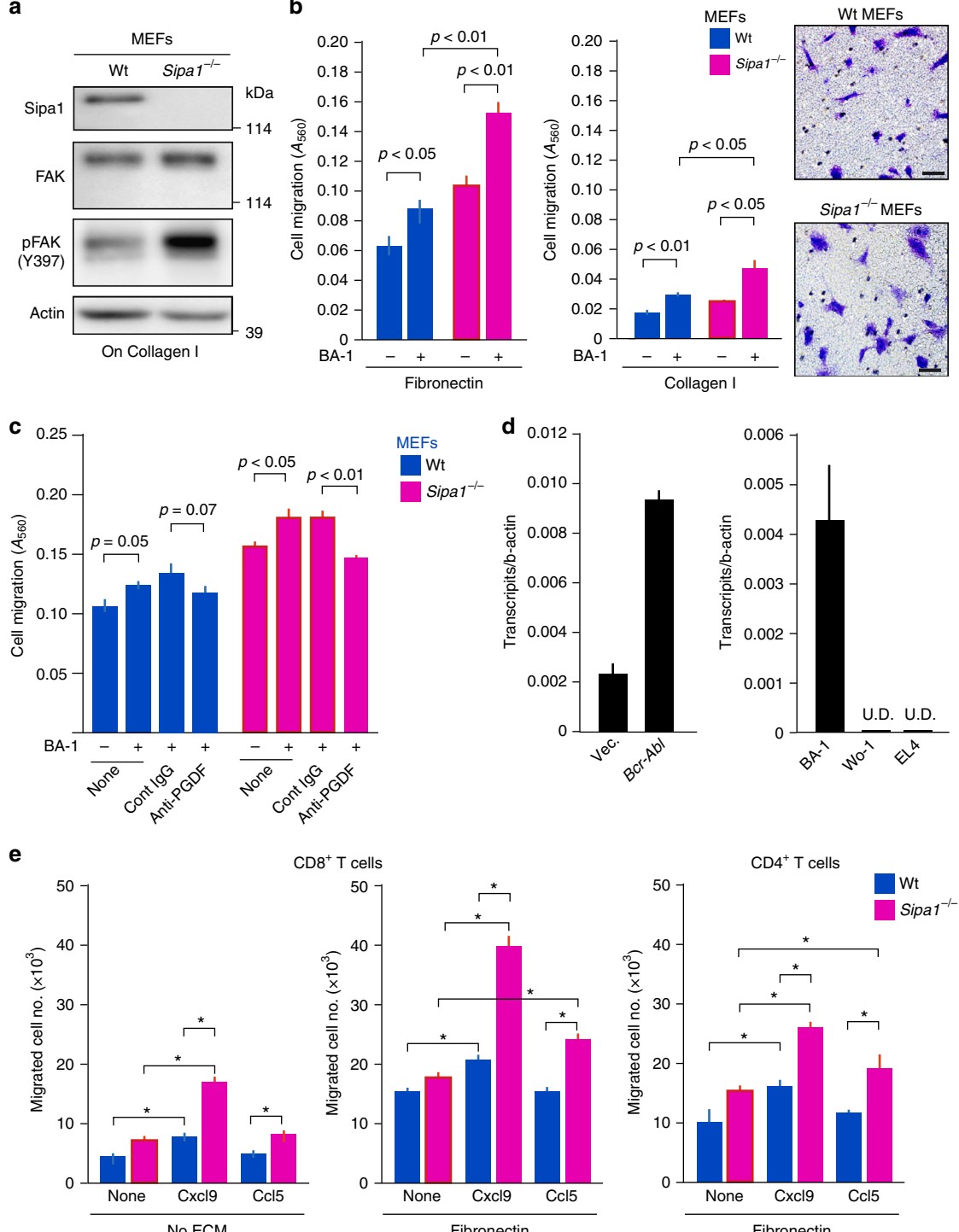

**Fig. 8** $Sipa1^{-/-}$ MEFs show enhanced migration directed to $Bcr-Abl^+$ CML cells and $Sipa1^{-/-}$ T cells augmented chemotaxis in response to chemokines. **a** Wt and $Sipa1^{-/-}$ MEFs cultured on collagen I were lysed and immunoblotted with indicated antibodies. **b** Directed migration of Wt and $Sipa1^{-/-}$ MEFs to BA-1 CML cells were assessed using the Boyden chamber assay in the presence of fibronectin or collagen I. The means and SEs of quadruplicate culture are shown, and $p$ values were determined with two-tailed unpaired Student's $t$-test. The experiments were repeated three times with similar results. Pictures of crystal violet-stained migrated cells are also indicated. Scale bars, 50 μm. **c** In the Boyden chambers, anti-PDGFa antibody or isotype-matched control IgG was added to the lower wells at 20 μg/mL with BA-1 cells. The means and SEs of quadruplicate culture are shown, and $p$ values were determined with two-tailed unpaired Student's $t$-test. **d** Primary Lin⁻ BM cells were transduced with p210Bcr-Abl or control vector. Three days later, GFP⁺ cells were sorted, and $Pdgfa$ expression was assessed with quantitative PCR (left). Expression of $Pdgfa$ in BA-1, Wo-1, and EL4 cell lines was also examined (right). U.D., undetectable. **e** Chemotactic activity of sorted Wt and $Sipa1^{-/-}$ CD8⁺ T cells and CD4⁺ T cells that had been activated on the coated anti-CD3 antibody for 24 h were assessed in response to Cxcl9 and Ccl5 (100 ng/mL) using the Boyden chamber assay in the presence or absence of fibronectin. The means and SEs of triplicate determination are shown. The experiments were repeated at least two times with similar results. *$p < 0.01$ (two-tailed unpaired Student's $t$-test)

Supplementary Fig. 8a). In the Boyden chamber assay, $Sipa1^{-/-}$ MEFs showed significantly enhanced directed migration to $Bcr$-$Abl^+$ BA-1 cells across the membrane compared to Wt MEFs ($p < 0.01$, two-tailed unpaired Student's $t$-test); the effect was more prominent in the presence of fibronectin than collagen I (Fig. 8b). Such migrated $Sipa1^{-/-}$ MEFs showed markedly spread forms compared to Wt MEFs (Fig. 8b). The enhancement of the ECM-dependent directed migration, referred to as haptotaxis[35], of $Sipa1^{-/-}$ MEFs to BA-1 cells was significantly inhibited in the presence of anti-PDGFa or anti-PDGFRα antibody ($p < 0.01$, two-tailed unpaired Student's $t$-test) (Fig. 8c, Supplementary Fig. 8b). We found that $Bcr$-$Abl$ expression induced an increase in $Pdgfa$ expression in the primary BM HPCs; $Bcr$-$Abl^+$ BA-1 cells consistently exhibited significant $Pdgfa$ expression, whereas Wo-1 and EL4 cells did not (Fig. 8d). The results suggested the involvement of leukemia-derived PDGF in the accumulation and activation of $Sipa1^{-/-}$ MSCs inside $Bcr$-$Abl^+$ tumor tissue. Given that $Sipa1^{-/-}$ T cells are much more effective at rejecting $Bcr$-$Abl^+$ cells than Wt T cells in the $Sipa1^{-/-}$ environment, we also examined the chemotactic activity in response to chemokines. Activated $Sipa1^{-/-}$ T cells exhibited remarkably enhanced chemotaxis compared to Wt T cells in response to Cxcl9 and, to a lesser extent, Ccl5; the chemotaxis was also more prominent in the presence of fibronectin (Fig. 8e). In agreement with the relative dominance of $CD8^+$ T cells in the tumor tissue, $Sipa1^{-/-}$ $CD8^+$ T cells showed greater chemotactic activity than $Sipa1^{-/-}$ $CD4^+$ T cells (Fig. 8e). Altogether, these results suggest that $Sipa1^{-/-}$ MSCs exhibit enhanced migration directed to $Bcr$-$Abl^+$ cells and secretion of Cxcl9, which causes efficient recruitment of $Sipa1^{-/-}$ memory T cells to the vicinity of the tumor cells (Supplementary Fig. 9). Secretion of Ccl5 by such recruited $CD8^+$ T cells may further amplify the T-cell recruitment inside the $Bcr$-$Abl^+$ tumor tissue.

## Discussion

Sustained expression of $Bcr$-$Abl$ in CML-initiating cells is essential for the maintenance and progression of CML disease[36]. Although massive expansion and accumulation of CML cells representing the differentiated progenies may be controlled by TKIs, CML-initiating cells are much less sensitive to TKIs than the majority of CML cells, partly because Bcr-Abl kinase activity alone is not sufficient to confer the leukemic stem cell activity[15,17,37,38]. Persisting CML-initiating cells as well as mutations in $Bcr$-$Abl$ kinase domain leading to drug resistance remain a potential risk for disease recurrence[39,40].

In the current study, we unexpectedly found that the primary $Bcr$-$Abl^+$ HPCs failed to develop CML disease in $Sipa1^{-/-}$ B6 mice, although they caused lethal CML with massive leukemic expansion of $Bcr$-$Abl^+$ cells in Wt B6 mice as expected. This finding may be reminiscent of $PD$-$1^{-/-}$ mice that show strong resistance to certain otherwise lethal tumors expressing $PD$-$L1^7$. The resistance of $Sipa1^{-/-}$ mice was abrogated by prior γ-ray irradiation and therefore overlooked in the standard procedures for the mouse CML model using BMT[34]. The effect was not attributable to the failure of the $Sipa1^{-/-}$ BM microenvironment to support proper engraftment of transferred $Bcr$-$Abl^+$ HPCs because the $Bcr$-$Abl^+$ HPCs did proliferate and differentiate until about 10 days in $Sipa1^{-/-}$ BM comparably to Wt BM. Rather, $Sipa1^{-/-}$ recipients actively rejected the $Bcr$-$Abl^+$ cells later, demonstrating a spontaneous cure with the complete elimination of CML-initiating cell activity. Genetic as well as cell depletion studies revealed that T cells of the $Sipa1^{-/-}$ host, including both $CD4^+$ and $CD8^+$ cells, are essential for the rejection of $Bcr$-$Abl^+$ HPCs, and the mice that once rejected $Bcr$-$Abl^+$ HPCs apparently established the memory state. In agreement with the findings, a much greater number of T cells were detected around $Bcr$-$Abl^+$ cells in $Sipa1^{-/-}$ BM compared to those in Wt BM.

The resistance of $Sipa1^{-/-}$ mice was apparently selective for $Bcr$-$Abl^+$ CML in that $Sipa1^{-/-}$ mice were comparably susceptible to other leukemia cells without $Bcr$-$Abl$ expression relative to Wt mice. Although $Bcr$-$Abl^+$ CML expressed GFP in the current model, $Sipa1^{-/-}$ mice were no more resistant than Wt mice to T-ALL cells similarly expressing GFP, and thus it was unlikely that GFP was involved in the immune rejection. The $Bcr$-$Abl$ fusion gene may create new antigenic epitopes, or neo-antigens, via a codon slip, and Bcr-Abl-specific T cells are detected in the CML patients[18,41,42]. It has also been reported that T cells specific for proteinase 3 (P3), a self antigen that is abundantly and exclusively expressed on normal myeloid cells, can emerge in CML patients under allogeneic BMT or interferon-α therapy but rarely in untreated patients; notably, the generation of these cells correlated with disease remission[19]. $Sipa1^{-/-}$ mice are prone to systemic autoimmunity[28], and an intriguing possibility is that the $Sipa1^{-/-}$ host favors the development of such T cells specific for self antigens expressed on robustly expanding CML cells. In either case, $Bcr$-$Abl^+$ CML cells can be potentially immunogenic for T cells, and crucial tumor-rejection antigens involved in a current model remain to be verified.

Current results, however, also revealed that $Sipa1^{-/-}$ T cells are ineffective for rejecting $Bcr$-$Abl^+$ HPCs in the Wt host environment. Thus, analysis using BM chimeric mice indicated that neither <Wt BM into $Sipa1^{-/-}$>mice nor <$Sipa1^{-/-}$BM into Wt>mice showed strong CML resistance comparable to that of <$Sipa1^{-/-}$ BM into $Sipa1^{-/-}$>mice, suggesting that the combination of $Sipa1^{-/-}$ T cells and $Sipa1^{-/-}$ nonhematopoietic cells is essential for the complete rejection. $Sipa1^{-/-}$, but not Wt, mice also rejected subcutaneous $Bcr$-$Abl^+$ tumors completely, suggesting that the required $Sipa1^{-/-}$ nonhematopoietic cells do not need to be specialized hematopoietic stroma cells. Histological analysis revealed a remarkable accumulation of vimentin-positive MSCs in concordance with abundant penetration of T cells inside the $Bcr$-$Abl^+$ tumor tissue of the $Sipa1^{-/-}$ host. In the Wt host, MSCs were scarcely detected in the tightly packed tumor tissue, and T cells remained at the peripheral margins of the tissue, if any. The results are consistent with the notion that infiltration of T cells in the center of tumor tissue is crucial for tumor regression[5], and they suggest that the accumulation and/or persistence of $Sipa1^{-/-}$ MSCs around the $Bcr$-$Abl^+$ tumor cells plays an important role in the intratumor T cell recruitment. Notably, γ-ray radiation profoundly damages not only T cells but also MSCs[43].

Intratumor $Sipa1^{-/-}$ MSCs showed significant activation of selective gene sets, including mesenchymal lineage genes, ECM and integrin genes, and FGF-related genes, compared to those in Wt mice. Because Sipa1 in the cytosol binds the nuclear chromatin modifying factor Brd4, which regulates the expression of a panel of ECM and the related receptor genes[44,45], the increase in nuclear Brd4 in $Sipa1^{-/-}$ MSCs may underlie the effects[31,32]. Enhanced Rap1 activation linked to protein tyrosine kinase (PTK) receptors in Sipa1 deficiency may also contribute to MSC activation via augmented MAPK signaling[22,24]. We found that $Bcr$-$Abl$ expression in the primary HPCs induces $Pdgfa$ expression. MSCs constitutively express PDGFR, and PDGF is a principal factor inducing the activation of MSCs, including directed cell migration via chemotactic ECM cues, called haptotaxis[35,46]. $Sipa1^{-/-}$ MEFs showed markedly enhanced basal FAK activation and haptotaxis directed to $Bcr$-$Abl^+$ BA-1 cells in an ECM-dependent manner compared to Wt MEFs, and the effect was inhibited in the presence of either anti-PDGF or anti-PDGFR antibody. Therefore, we suggest that tumor-derived PDGF may be involved, at least partly, in the accumulation and activation of

*Sipa1*$^{-/-}$ MSCs in the center of tumor tissue. The role of resident MSCs in tumor tissue is controversial. Certain human cancers such as breast cancers are often associated with persistent "reactive stroma" consisting of fibroblasts showing characteristic phenotypic transition, including strong α-smooth muscle actin (SMA) expression[47]. Such cancer-associated fibroblasts that develop under the influence of transforming growth factor (TGF)-β may show tumor-promoting and/or immunosuppressive activity, often correlating with poor patient prognosis[3,48,49]. However, resident fibroblasts in tumor tissue have been reported to potentially be able to inhibit tumor progression[50]; for instance, blocking TGF-β signaling in MSCs reveals the tumor-suppressive activity[51].

Importantly, *Sipa1*$^{-/-}$ MSCs, but rarely Wt MSCs, in *Bcr-Abl*$^+$ tumor tissue preferentially and strongly expressed chemokine genes such as *Cxcl9 (Mig)* targeting memory T cells with minimal proinflammatory chemokines targeting myeloid cells. A recent study has suggested that tissue MSCs may develop into distinct functional subsets depending on the nature of microenvironmental stresses[52]. It remains to be seen whether the *Sipa1*$^{-/-}$ MSCs preferentially producing Ccl9 represent a unique functional subset or reflect the specific mode of MSC activation in the absence of Sipa1. Furthermore, activated *Sipa1*$^{-/-}$ T cells, in particular CD8$^+$ cells, exhibited markedly enhanced chemotactic activity in response to Cxcl9, most likely due to the increased Rap1-mediated inside-out activation of migratory integrins[23]. We found that *Sipa1*$^{-/-}$ memory CD8$^+$ T cells in tumor tissue further produce Ccl5 (Rantes), which shows potent TCR costimulatory activity for CD4$^+$ T cells in addition to T-cell chemotactic activity[53]. The results revealed a potentially important role of MSCs in recruiting memory T cells in the center of tumor tissue to ensure their direct encounter with tumor cells for the immune effects. The mechanism is remarkably potentiated in the *Sipa1*$^{-/-}$ host, leading to the complete immunological eradication of CML; however, it remains rather latent in the Wt host. Most recently, cooperation between T cells and nonhematopoietic stroma cells leading to the rejection of a head and neck cancer cell line was reported in mice with combined deficiency of *Toll-like receptors 3, 7,* and *9*[54].

Large-scale cancer patient analyses have indicated that specific patterns of immune activation within tumor tissues are associated with a better prognosis for cancer patients. These patterns include an abundance of memory T cells as well as molecular signatures related to T-cell chemotaxis, cytotoxicity, and cell adhesion[5]. Notably, the location of T cells in the center and invasive margin of tumor tissue is significantly correlated with better prognosis[55,56]. Therefore, in clinical settings, recruitment of memory T cells inside the tumor tissue ensuring the functional access of immune T cells to tumor cells may be important for suppressing tumor progression. Our current study suggests that the coordinated interplay between MSCs and immune T cells inside tumor tissue has a strong potential for leading to a cure for CML disease. Although it remains to be proven whether properly accessed immune T cells can directly affect TKI-resistant CML-initiating cells and *Bcr-Abl* mutant cells, current results may provide a novel clue for radically controlling human CML.

## Methods

**Mice.** C57BL/6 (B6) mice were purchased from Charles River Laboratories Inc. (Kanagawa, Japan) and SLC Inc. (Shizuoka, Japan). *Rag2*$^{-/-}$ mice were obtained from Central Institute of Experimental Animals (Kanagawa, Japan). *CD3ε*$^{-/-}$ and *μMT*$^{-/-}$ mice were provided by Dr. S. Fagarasan (RIKEN Center for Integrative Medical Science, Kanagawa, Japan) and Dr. M. Reth (Albert Ludwigs Universität Freiburg, Freiburg, Germany), respectively. All these mutant mice had a B6 genetic background. *Sipa1*$^{-/-}$ mice have been reported previously[26] and crossed with B6 mice for more than 15 generations. Enhanced GFP (EGFP)-Sipa1 reporter (*EGFP; Sipa1* KI) mice were generated as a custom order by Unitech (Kashiwa, Chiba,

Japan). In brief, the *Sipa1* chromosomal gene was isolated from bacterial artificial chromosome clones of C57BL/6 mice, and its exon 4 was replaced by a coding sequence for *EGFP* and *loxp-neo-loxp* genes. The targeting vector containing the diphtheria toxin A fragment upstream exon 9 was transfected into embryonic stem (ES) cells (clone: Bruce 4) derived from C57BL/6 mice. The ES clones carrying the targeted allele were introduced into host embryos, and the chimeric mice were crossed with the *CAG-Cre* transgenic mice. In all experiments, male and female mice of 6–8-week old were used, and the numbers of mice used in experiments were indicated in figure legends. All mice were maintained in specific pathogen-free conditions at the Center for Experimental Animals of Kyoto University, and all animal experiments were performed strictly according to the institutional guidelines by Kyoto University Animal Ethics Committee.

**Cells and cultures.** The BA-1 cell line was established from the bone marrow cells (BMCs) of young *Sipa1*$^{-/-}$ mice by infection with *pMSCV-IRES-GFP* retroviral plasmid (*MIG*) containing human p210 Bcr-Abl. The Wo-1 cell line was established from thymic T-cell acute lymphoblastic leukemia (T-ALL) developed by the BMT of HPCs transduced with *MIG* containing *C3G-F*[57], and the T-cell leukemia cell line, EL4, was purchased from American Type Culture Collection. The cells were cultured with complete BXH-2 medium supplemented with 10% fetal calf serum (FCS)[26]. The leukemic cell lines were authenticated by the gene expression (*C3G-F*) and FACS analysis of markers (CD3, CD4). Mouse embryonic fibroblasts (MEFs) were obtained from the embryos of Wt and *Sipa1*$^{-/-}$ mice on day 15.5 of gestation and maintained with Dulbecco's modified Eagle medium (DMEM) containing 10% FCS, unless otherwise specified. The MEFs were checked for the expression of Sipa1 with immunoblotting. All cells were tested for mycoplasma contamination.

**Retrovirus infection and cell transfer.** Bone marrow cells (BMCs) from 4- to 6-week-old B6 mice pretreated with 150 mg/kg 5-fluorouracil (5-FU; Kyowa Hakko Kogyo, Tokyo, Japan) were depleted of lineage marker-positive (Lin$^+$) cells using a cocktail of antibodies (anti-CD3, anti-CD4, anti-CD8, anti-B220, anti-Gr-1, anti-Mac-1, and anti-Ter119 antibodies; BD Bioscience, San Jose, CA, USA) and anti-rat IgG-coated magnetic beads (Supplementary Table 1) (Dynabeads M-450; Dynal, Oslo, Norway). The resulting Lin$^-$ BMCs enriched for HPCs were infected with *MIG/p210Bcr-Abl* retrovirus as previously reported[58]. Briefly, the cells were infected with retroviral supernatants by spinoculation and cultured in complete BXH-2 medium containing 10 ng/mL IL-6, 10 ng/mL IL-11, 10 ng/mL Flt-3 ligand, and 50 ng/mL stem cell factor (Genzyme, Minneapolis, MN) for 48 h. Infection efficiency was 30–35%. The infected cells were injected intravenously or subcutaneously into mice at $1.5 \times 10^4$ cells per mouse. For subcutaneous injection, the GFP$^+$ HPCs were placed in Matrigel matrix (Corning, Tewksbury, MA, USA) to prevent rapid cell diffusion.

**BM chimeras.** BMCs from Wt or *Sipa1*$^{-/-}$ mice were injected intravenously into Wt or *Sipa1*$^{-/-}$ mice irradiated with 13-Gy γ-ray (6.5 Gy twice with a 4-h interval) at $2 \times 10^7$ cells per mouse. Chimeric rates in the procedures were more than 95%. Chimeric mice were rested for 4–6 months before the challenge with *Bcr-Abl*$^+$ HPCs.

**Flow cytometry.** Multicolor flow cytometric analysis was performed with FACS Canto (BD Bioscience, San Jose, CA, USA), and the data were analyzed with FlowJo (Tree Star). Cells were Fc-blocked (CD16/CD32) and stained with antibodies (Supplementary Table 1). Propidium iodide or Ghost dye$^{TM}$ Violet 450 (Tonbo Biosciences) was used to exclude dead cells. Intracellular staining for Ccl5 was performed with cells fixed and made permeable with IC Fixation Buffer and Permeabilization Buffer (eBioscience) according to the manufacturer's instructions.

**Immunostaining.** Femoral bones or subcutaneous tumor tissues were fixed in 2% paraformaldehyde for 2 h at room temperature and left in 30% sucrose at 4 °C overnight before embedding in optimum cutting temperature compound (Sakura, Torrance, CA). Serial frozen sections were blocked with 5% donkey serum in phosphate-buffered saline (PBS) overnight followed by immunostaining with antibodies (Supplementary Table 1) as before[59]. Stained tissues were mounted with Mowiol 4–88 (Calbiochem), and images were acquired with the use of fluorescence microscopy (Axiovert 200M) equipped with an AxioCam MRm Fluar 2.5×/0.12 or 5×/0.25 numerical aperture objective lens (Carl Zeiss) and analyzed with Axio-Vision version 4.6 (Carl Zeiss). Digital images were processed using Adobe Photoshop CS2 (Adobe). For immunostaining of MEFs, cells were grown on coverslips (Matsunami Glass Ltd., Japan), fixed with 4% paraformaldehyde for 10 min at room temperature, permeabilized with 0.1% Triton X-100 for 5 min, blocked with 5% BSA/PBS for 1 h, and incubated with primary antibodies for 1 h at room temperature followed by secondary antibodies and DAPI for 30 min. The antibodies used were anti-phospho-FAK (ab81298; Abcam, Cambridge, UK, at 1:1000 dilution), Alexa Fluor 568 Phalloidin (Invitrogen), and Alexa Fluor 488. Images were obtained using the BZ-X700-All-in-One fluorescence microscope (Keyence).

**Antibody-mediated cell depletion.** To specifically deplete NK cells in vivo, anti-NK1.1 antibody (PK136) was injected intraperitoneally at 200 μg per mouse twice

per week before and once per week after *Bcr-Abl*[+] HPC injection. For CD4[+] and CD8[+] T-cell depletion, anti-CD4 (GK1.5) and anti-CD8 (2.34) antibodies were injected at 150 µg per mouse, respectively, 2 days before and twice per week after *Bcr-Abl*[+] HPC injection. Control groups were treated with corresponding isotype-matched control IgG. The procedures were confirmed to have depleted more than 95% of the corresponding cell populations.

**Tumor measurement and isolation of tumor-infiltrating cells**. The sizes of subcutaneous tumors were measured in two dimensions, and the volumes were assessed by the following formula: tumor volume $= l/2 \times w^2$. Tumor tissues (5 mg) were cut into small pieces in 1 mg/mL Collagenases IV and DNase I in 2% FCS RPMI-1640 medium. After 60 min of incubation with gentle shaking at 37 °C, the cells were strained through a 40-µm nylon mesh to produce a single-cell suspension. To isolate mesenchymal stroma cells (MSCs), the cell suspensions were stained with anti-GFP and anti-CD45-BV510 (30-F11; BD Bioscience), anti-TER119-BV510 (TER119; BD Bioscience), anti-CD31-APC (390; BioLegend), and anti-PDGFRα (APA5; eBioscience) antibodies at 1 µg/mL. The GFP[−] CD45[−] Ter119[−] CD31[−] PDGFRα[+] cell population was sorted with FACS Aria III (BD Bioscience). In some experiments, anti-PDGFRα was omitted to ensure enough cell numbers. Contamination of GFP[+] cells and CD31[+] ECs in the MSC population was less than 0.5% and 0.1%, respectively.

**DNA microarray and qPCR analysis**. RNA was extracted from the sorted GFP[−] CD45[−] Ter119[−] CD31[−] cell populations and subjected to DNA microarray analysis (TaKaRa Bio Inc., Siga, Japan). Gene expression profiling was performed using total RNA and microarray (SurePrint G3 Mouse Gene Expression v2 8 × 60 K Microarray Kit, Agilent) according to manufacturer's protocol. Probes with confident expression signals were selected using flags produced by Feature Extraction software: IsSaturated = 0, IsFeatNonUnifOL = 0, IsBGNonUnifOL = 0, IsPosAndSignif = 1 and IsWellAboveBG = 1. All selected probes corresponding to ~19,000 genes were subjected to Gene Set Enrichment Analysis using expression log ratio $\log_2$ (KO/WT) as rank metrics. The analysis was done with following settings: preranked analysis mode of GSEA software (version 2.2.4, Broad Institute) and gene sets from category Hallmark and C2, which were prepared in MSigDB (Broad Institute). Quantitative PCR was performed with SYBR Green I Master mix (Roche) on a LightCycler 480 (Roche). Transcripts of *b-actin* were used as an internal control. Primer sequences for genes are listed in the Supplementary Table 2.

**Cytokine array analysis**. Cytokine levels in tumor tissue supernatants were measured using Mouse Cytokine Antibody Array, Panel A (R&D Systems), according to the manufacturer's instructions. Samples were analyzed as singlets using a LAS-4000 mini instrument with Image Reader Las-4000 mini and Multi Gage V3.2 software (Fujifilm).

**Cell migration assay**. The haptotaxis assay was performed using the CytoSelect™ 24-Well Cell Haptotaxis Assay Kit (8 µm, fibronectin-coated or collagen-coated, Cell Biolabs, Inc.) with 8.0-µm pore membrane inserts coated with fibronectin or collagen I, according to the manufacturer's procedures. Briefly, MEFs ($1 \times 10^4$ cells) prestarved of serum for 1 h were applied to the upper wells in the presence or absence of $2 \times 10^6$ BA-1 cells in the lower wells in DMEM containing 0.1% FCS, followed by incubation at 37 °C for 4 h. When indicated, 20 µg/mL anti-PDGFa (Upstate), anti-PDGFRα (R&D Systems), or isotype-matched control IgG was added to the lower wells. The outer parts of the inset membranes were stained with crystal violet, and OD 560 nm was measured with ARBO™ X 2030 Multilabel Reader (PerkinElmer). The chemotaxis assay was performed using 24-well plates and 6.5-mm Transwell with 5.0-µm pore polycarbonate membrane inserts (Corning, NY, USA). Inserts were coated with 10ug/mL fibronectin (Invitrogen) for 1 h and dried. CD4[+] and CD8[+] T cells were isolated with autoMACS Pro Separator (Miltenyi Biotec, Bergisch Gladbach, Germany) and added to the upper chamber ($5 \times 10^5$ cells) in 1% BSA RPMI-1640. The lower chambers were filled with 1% BSA RPMI-1640 containing Cxcl9 or Ccl5 (PeproTech, Rocky Hill, NJ, USA) at 100 ng/mL. After 3 h of incubation in 5% $CO_2$ at 37 °C, the cells that migrated into the lower chambers were counted by FACS Canto (BD Bioscience).

**Immunoblotting**. MEFs were lysed with lysis buffer (150 mM NaCl, 25 mM Tris-HCl, 1%NP-40, 1% Na deoxycholate, 0.1% SDS) and blotted with anti-Sipa1[26], anti-FAK (ab40794;Abcam, Cambridge, UK), or anti-phospho-FAK (Y397;Abcam, Cambridge, UK) antibody at 1:1000 dilution. Uncropped original scan images are shown in Supplementary Fig. 10.

**Statistical analyses**. Statistical analyses were performed using the unpaired two-tailed Student's *t*-test and one-way ANOVA with Bonferroni post hoc test. Spearman's rank test was used to determine correlations; $p < 0.05$ indicated a significant difference. In animal experiments, it was not feasible to make an assumption of sample sizes based on the log-rank statistic, because we have never observed the development of lethal CML in *Sipa1*[−/−] mice for observation period of 100 days (i.e., zero event numbers). Age-matched mice kept in the same animal

room were used with no particular randomization, and no mice were excluded from the analysis. The investigators were not blinded for group allocations.

**Data availability**. The DNA Microarray data were deposited in Gene Expression Omnibus (GEO) repository, NCBI (GSE108002, https://www.ncbi.nlm.nih.gov/geo/query/acc.cgi?acc=GSE108002). The authors declare that all the other data supporting the findings of this study are available within the article and its Supplementary Information files and from the corresponding author upon reasonable request.

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

## Acknowledgements

We thank Dr. R. Inagaki (Genome Research Institute, Sumitomo Dainippon Pharma Co., Ltd.) for helping with the data analysis of DNA microarray, Drs. Y. Hamazaki and M. Sekai (Center for iPS Research and Application, Kyoto University) for advising on immunostaining, and Dr. S. Morita (Graduate School of Medicine, Kyoto University) for helping statistic analysis. This work was supported by a grant-in-aid for Scientific Research on Innovative Areas, MEXT, Japan (to N.M.) and partly by Sumitomo Dainippon Pharma Co., Ltd.

## Author contributions

Y.X. performed the majority of tumor experiments in vivo, BM immunostaining, and PCR analysis. S.I. performed the cellular and histochemical analysis of subcutaneous tumors. K.S. performed the cell depletion analysis in vivo and chemotactic assay. R.Y. performed the cell depletion analysis in vivo. H.T. performed leukemia experiments in vivo and the haptotaxis assay with MEFs. N.M. planned the overall experiments and wrote the paper.

## Additional information

**Competing interests:** This work was supported in part by research funding from Sumitomo Dainippon Pharma Co., Ltd.

