## [Peer Review File · Nature Communications]

Reviewers' comments:

Reviewer #1 (Remarks to the Author):

The authors report on a novel role of Sipa1 in the elimination of Bcr-Abl+ CML in a mouse model. They showed that mesenchymal stroma cells (MSCs) in Sipa1 mice have increased migration to Bcr-Abl+ CML, in both bone marrow and extramedullary tissues, which produce Cxcl9 that results in increased accumulation of memory T cells that are associated with elimination of the Bcr-Abl+ CML from Wt B6 mice. This is a well-established model that was first used to establish causality of bcr-abl in initiating CML by the Witte group in 1993. The novel role of Sipa1 in the current study stems from Xu et al's initial observation that CML is eliminated in nonirradiated Sipa1-/- recipients but not in irradiated animals. The conclusion that T cells are critical for the elimination of CML is not novel, since this is clinically well-established. However, this model, where the authors have previously shown that Sipa1-/- leads to autoimmunity, could be useful to understand mechanisms of resistance of CML-initiating cells to elimination by T cells specific for self/leukemia-associated antigens.

1. Aside from minor syntactical and spelling errors, the manuscript is well written and the data are compelling. Some questions remain, however, regarding additional control experiments, potential alternative interpretations of the data, and the supposition of potential clinical significance where TKI-resistance has been observed in CML stem cell cells.
2. In the introduction, the authors state that CML stem cells escape immune surveillance (lines 57-59). While this is true in the host, CML can be eliminated by allogeneic T cells from healthy donors, and antigen-specific CD8 T cells have been shown to contribute to long-term complete remission (Molldrem et al, Nat Med, 2000). How do the authors interpret the significance of their findings to the clinical setting?
3. An alternative explanation for the results in Figure 2 could be that the irradiation used in this model affects capacity for Bcr-Abl+ cell self-renewal and/or lineage differentiation.
4. The conclusion in lines 142-144 seems to be overstated since quant beads were not used to quantify expression. Moreover, the results seem to show two populations of memory T cells – was this consistent in other experiments? Could this be relevant to the role of the memory T cells in anti-CML activity?
5. How do the authors control for contamination of residual Bcr-Abl+ cells in the gene expression results in Figure 7/Supplementary Figure 4?
6. Single cell analysis of CML patients treated with TKI has shown a role for altered expression of TGF-beta, NF-kB, WNT/beta-catenin, and JAK-STAT in the expansion of a rare population of quiescent CML stem cells (Giustacchini et al, Nat Med, June, 2017). Were any alterations in the expression of these cytokines observed in the Sirp1-/- model? If not, how does this impact the relevance of this model and the observations reported in these experiments to CML?

Reviewer #2 (Remarks to the Author):

Review of "A potent immune mechanism eradicating chronic myelogenous leukemia-initiating cells uncovered by Sipa1 deficiency" by Xu et al.

This manuscript shows that the ability of CML to expand and cause lethal disease in an animal model can be limited by tumor extrinsic factors controlled by the Sipa1 gene. The authors show that the cell responsible for this tumor extrinsic control of CML is radio sensitive. The CML cells (retroviral BCR-ABL IRES GFP+ transduced HSC) engraft into recipients BM but disappear between

12-15 days. Using genetic and antibody depletion studies the authors show that resistance to CML is dependent on both CD4 and CD8 T cells but not B or NK cells. Using chimeras the authors show that while the main protective effect is hematopoietic derived, that a small amount of protection may be due to stromal components. The authors show that Sip1 is expressed in a variety of immune and stromal components. The authors then go on to show that there are changes in the MSC compartment of Sip1^{-/-} mice produce T cell recruiting cytokines to a higher degree and that T cells from these animals migrate more efficiently to Cxcl9 and Ccl5 when competed to Sip1 wt T cells.

The experiments are very well done and the results are clear. The histology of the bones to identify CML cells is beautiful. Overall this manuscript shows a very exciting new potential pathway by which immune cells as well as stromal cells may control tumors.

There are several questions that are critical remaining.

1. The effect of the MSC on the phenomenon is not clear in this reviewer's mind. This interpretation is based on the chimera data shown in figure 3 WT into Sip1^{-/-} chimera (red dotted line). The effect is, while statically significant, somewhat weak. With 12 animals in this group a 20% survival (line 152 of the text) would be 2 animals. This is a small effect size and careful statistics need to be applied. Further, in supplemental figure 1, this effect appears to be lost when using the BA-1 CML cell line. In supplemental figure 1b it appears that the surviving WT into Sip1^{-/-} chimeras is around 5% which would only be 1 animal out of 22. It is difficult to evaluate well due to the scale and lack of statistics in the supplemental figure. A power calculation should be done to determine in this study how many animals would be needed to detect the 20% difference seen between WT into WT and WT into Sip1^{-/-} chimera groups. 12 mice is likely too small to detect the difference. How many times the experiment was repeated? The number of replicates of each experiment must be listed in the figure legends for all the figures.

2. The presence (quantity) and phenotype of T cells in the marrow of the Sip1^{-/-} mice rejecting tumor must be identified. Is there truly a difference in recruitment of T cells to the marrow of the Sip1^{-/-} mice or is the increased survival due to more active T cells? The gene expression data would lead one to predict that there would be an influx of CD4 and CD8 T cells into the marrow of the Sip1^{-/-} mice, is this true?

3. A major claim in the paper is that this immune mediated rejection of tumor in Sip1^{-/-} mice is CML specific. This is based on the comparison of primary BCR-ABL transduced and BA-1 cells which are rejected to Wo-1 (T-ALL) and EL4 (Lymphoma). A major confounding factor in this interpretation is that both the primary transduced cells and BA-1 are made with an IRES-GFP while Wo-1 and EL4 lack GFP. It is highly likely that the rejection of these lines is due to recognition and targeting of GFP by the T cells. An experiment should be conducted to either express GFP in Wo-1 and EL4 OR remove GFP from the BCR-ABL construct and repeat the experiments.

Overall in summary the paper is very exciting and will add greatly to the field if the above issues are clarified.

Response to Reviewers' comments:

Reviewer #1 (Remarks to the Author):

The authors report on a novel role of Sip1 in the elimination of Bcr-Abl+ CML in a mouse model. They showed that mesenchymal stroma cells (MSCs) in Sip1 mice have increased migration to Bcr-Abl+ CML, in both bone marrow and extramedullary tissues, which produce Cxcl9 that results in increased accumulation of memory T cells that are associated with elimination of the Bcr-Abl+ CML from Wt B6 mice. This is a well-established model that was first used to establish causality of bcr-abl in initiating CML by the Witte group in 1993. The novel role of Sip1 in the current study stems from Xu et al's initial observation that CML is eliminated in nonirradiated Sip1-/- recipients but not in irradiated animals. The conclusion that T cells are critical for the elimination of CML is not novel, since this is clinically well-established. However, this model, where the authors have previously shown that Sip1-/- leads to autoimmunity, could be useful to understand mechanisms of resistance of CML-initiating cells to elimination by T cells specific for self/leukemia-associated antigens.

1. Aside from minor syntactical and spelling errors, the manuscript is well written and the data are compelling. Some questions remain, however, regarding additional control experiments, potential alternative interpretations of the data, and the supposition of potential clinical significance where TKI-resistance has been observed in CML stem cell cells.

We appreciate the accurate general comments by the reviewer, to which we accordingly responded as below. We have also polished the English with the aid of a native English-speaking editor.

2. In the introduction, the authors state that CML stem cells escape immune surveillance (lines 57-59). While this is true in the host, CML can be eliminated by allogeneic T cells from healthy donors, and antigen-specific CD8 T cells have been shown to contribute to long-term complete remission (Molldrem et al, Nat Med, 2000). How do the authors interpret the significance of their findings to the clinical setting?

Thank you for the excellent comment. We overlooked this important paper, which indicated that P3, a “self-antigen” that is abundantly and exclusively expressed in normal myeloid cells, can be a potential tumor antigen in CML. Notably, the development of P3-specific T cells is detected only after allogeneic BMT or IFN- α therapy but rarely in untreated or chemotherapy-treated CML patients, and it is correlated with the disease remission by the therapies. Therefore, it seems possible that certain immunotherapies promote the breakdown of self-tolerance for potential T-cell immunity against CML cells. This is in contrast to T cells specific for “neoantigens” from Bcr-Abl fusion protein, which may develop in many untreated CML patients irrespective of disease progression (Rusakiewicz et al., Cancer Immunol Immunother 2009). We previously reported that Sip1^{-/-} mice often develop overt autoimmunity, suggesting a defect in sustaining the self-tolerance (Ishida et al., Immunity, 2006), and as such, the involvement of potential self antigens such as P3 in

the CML resistance of the Sip $1^{-/-}$ host is a good possibility. Identification of specific antigens responsible for CML rejection in our model is certainly within the scope of our next step studies. We have discussed the important point in the revised text (p.17), citing the indicated reference (Ref. 19).

3. An alternative explanation for the results in Figure 2 could be that the irradiation used in this model affects capacity for Bcr-Abl+ cell self-renewal and/or lineage differentiation.

This is a crucial point and was also our concern. We believe the results in Fig. 2 indicate that Bcr-Abl⁺ HPCs properly home to the BM and initiate comparable proliferation in the Wt and Sip $1^{-/-}$ hosts, at least until about 10 days after cell transfer. Based on the comment, we also compared the differentiation profiles of Bcr-Abl⁺ HPCs in the Wt and Sip $1^{-/-}$ hosts, and we found that the lineage differentiation patterns were also largely comparable. The results are presented in **Supplementary Fig. 1**. Therefore, we concluded that the resistance of unirradiated Sip $1^{-/-}$ mice was due to active host reactions rather than failure of supporting Bcr-Abl⁺ HPC stem/progenitor activity. Since radioresistance of the hematopoietic niche function is well established, the radiosensitivity of the CML resistance seems to be attributed to the effects of γ -ray on the active host reactions (MSCs and/or T cells). The point is discussed in the revised text (p.16).

4. The conclusion in lines 142-144 seems to be overstated since quant beads were not used to quantify expression. Moreover, the results seem to show two populations of memory T cells – was this consistent in other experiments? Could this be relevant to the role of the memory T cells in anti-CML activity?

We agree with the comment. To ensure that the intensities of GFP reflect the actual expression levels of Sip 1 protein, we examined the Sip 1 protein expression in total spleen cells with intracellular staining and compared the expression levels among three fractions with different GFP intensities, GFP^{low}, GFP^{med}, and GFP^{high}. As indicated in figure A below, the mean fluorescence intensities (MFIs) of intracellular Sip 1 showed a fairly good correlation with the GFP intensities. With this in mind, we reanalyzed the naïve/memory phenotypes of CD4⁺ T cells in three GFP fractions. As also shown in figure B below, the naïve (CD62L^{high} CD44^{low}) CD4⁺ T cells were mostly confined to the GFP^{low} fraction, whereas memory (CD62L^{low} CD44^{high}) CD4⁺ T cells were distributed throughout three fractions, suggesting that Sip 1 expression increases progressively as CD4⁺ T cells develop into memory state. This suggestion is in agreement with our previous report that Sip 1 expression is transcriptionally activated after antigen stimulation (Ishida et al., PNAS 2003). These points are mentioned in the revised text (p. 8 and legend for Fig. 3). It is considered that the potential of cell adhesion and migration is markedly increased in the antigen-primed T cells, and therefore we presume that the increase in Sip 1 meets the requirement for much tighter regulation of these activities. Consequently, loss of Sip 1 expression in memory T cells may lead to robust activation of the cell migration potential compared to naïve T cells.

5. How do the authors control for contamination of residual Bcr-Abl⁺ cells in the gene expression results in Figure 7/Supplementary Figure 4?

We have sorted the MSCs (GFP⁻ CD45⁻ Ter119⁻ CD31⁻) with a rigorous gating condition using FACS Aria III, and, as indicated in the figure below, reanalysis of sorted cells showed practically undetectable contamination of GFP⁺ cells and CD31⁺ endothelial cells in both Wt and Sipa1^{-/-} mice (less than 0.5%). We also confirmed that mesenchymal lineage genes indicated in Fig. 7 were expressed minimally in sorted GFP⁺ tumor cells (0.1~0.01 times the transcripts of MSCs). In summery, we think it is very unlikely that possibly contaminated Bcr-Abl⁺ cells significantly contributed to the gene expression of MSCs. The point is mentioned in the revised legend for Fig. 7.

6. Single cell analysis of CML patients treated with TKI has shown a role for altered expression of TGF-beta, NF-kB, WNT/beta-catenin, and JAK-STAT in the expansion of a rare population of quiescent CML stem cells (Giustacchini et al, Nat Med, June, 2017). Were any alterations in the expression of these cytokines observed in the Sip1^{-/-} model? If not, how does this impact the relevance of this model and the observations reported in these experiments to CML?

This is another very important point. Based on the suggestion, we sorted the GFP⁺ CML cells from WT and Sip1^{-/-} mice with FACSARIA III on day 9 and compared the expression of the relevant genes that may affect their quiescence. As shown in the figure below, we found that the expression of *Tgfb*, *Tnfa*, and *Il-6* tended to be higher in the GFP⁺ CML cells from Sip1^{-/-} mice than those from WT mice, if not remarkably. This finding may imply that the residual CML cells in the Sip1^{-/-} host include more CML stem cells with increased quiescence than those in the Wt host. Nonetheless, we found that Sip1^{-/-} mice remained disease-free with undetectable Bcr-Abl⁺ LKS (progenitor) cells for more than 150 days, and more importantly their BM failed to transfer the CML into Wt mice even on a one-to-one basis (Fig. 2e), suggesting complete eradication of CML-initiating cells. Final proof for the immune eradication of quiescent CML stem cells resistant to TKI must await the identification of the relevant target antigens and the analysis of their expression on them. The point is discussed in the revised text (p. 21).

We appreciate the very constructive comments and suggestions by the reviewers.

Reviewer #2 (Remarks to the Author):

Review of “A potent immune mechanism eradicating chronic myelogenous leukemia–initiating cells uncovered by Sipa1 deficiency” by Xu et al.

This manuscript shows that the ability of CML to expand and cause lethal disease in an animal model can be limited by tumor extrinsic factors controlled by the Sipa1 gene. The authors show that the cell responsible for this tumor extrinsic control of CML is radiosensitive. The CML cells (retroviral BCR-ABL IRES GFP+ transduced HSC) engraft into recipients BM but disappear between 12-15 days. Using genetic and antibody depletion studies the authors show that resistance to CML is dependent on both CD4 and CD8 T cells but not B or NK cells. Using chimeras the authors show that while the main protective effect is hematopoietic derived, that a small amount of protection may be due to stromal components. The authors show that Sipa1 is expressed in a variety of immune and stromal components. The authors then go on to show that there are changes in the MSC compartment of Sipa1^{-/-} mice produce T cell recruiting cytokines to a higher degree and that T cells from these animals migrate more efficiently to Cxcl9 and Ccl5 when competed to Sipa1 wt T cells.

The experiments are very well done and the results are clear. The histology of the bones to identify CML cells is beautiful. Overall this manuscript shows a very exciting new potential pathway by which immune cells as well as stromal cells may control tumors.

We appreciate the encouraging general comment by the reviewer.

There are several questions that are critical remaining.

1. The effect of the MSC on the phenomenon is not clear in this reviewer's mind. This interpretation is based on the chimera data shown in figure 3 WT into Sipa1^{-/-} chimera (red dotted line). The effect is, while statically significant, somewhat weak. With 12 animals in this group a 20% survival (line 152 of the text) would be 2 animals. This is a small effect size and careful statistics need to be applied. Further, in supplemental figure 1, this effect appears to be lost when using the BA-1 CML cell line. In supplemental figure 1b it appears that the surviving WT into Sipa1^{-/-} chimeras is around 5% which would only be 1 animal out of 22. It is difficult to evaluate well due to the scale and lack of statistics in the supplemental figure. A power calculation should be done to determine in this study how many animals would be needed to detect the 20% difference seen between WT into WT and WT into Sipa1^{-/-} chimera groups. 12 mice is likely too small to detect the difference. How many times the experiment was repeated? The number of replicates of each experiment must be listed in the figure legends for all the figures.

Thank you for the important comment. First of all, the description “approximately 20%” was quite misleading, and we apologize for it. To be exact, the long-term survivors in

this experiment were three rather than two mice and therefore 25% as indicated in Fig. 3b. Nevertheless, we had the same concern as the reviewer on the data. Our initial pilot experiment indicated that three out of five <Wt into Sipa1^{-/-}> mice survived for at least 36 days, whereas all five <Sipa1^{-/-} into Wt mice> mice died before 27 days similar to <Wt into Wt> mice (please see the figure below), but the experiment was terminated at this point and could not be pooled with the data of Fig. 3b. We think the best and safer interpretation of the data would be that Wt hematopoietic cells (i.e. T cells) are significantly less effective in the CML rejection than Sipa1^{-/-} T cells even in the Sipa1^{-/-} nonhematopoietic environment, because at least the difference between this group and < Sipa1^{-/-} into Sipa1^{-/-}> mice should be convincingly significant (p<0.001). Given the data of BA-1 cells in Supplementary Fig. 1b (Fig. 2b in the revised manuscript), we agree that the difference between < Wt into Sipa1^{-/-}> and < Wt into Wt> mice may not be convincing at this experimental scale. Together with the susceptibility of < Sipa1^{-/-} into Wt > group being identical to < Wt into Wt > group, we only concluded that both hematopoietic and nonhematopoietic cell components are required for the optimal CML resistance seen in the Sipa1^{-/-} host in the revised text (p.18).

Also, as requested, we indicated the numbers of experiments in all presented figures.

2. The presence (quantity) and phenotype of T cells in the marrow of the Sipa-/- mice rejecting tumor must be identified. Is there truly a difference in recruitment of T cells to the marrow of the Sipa-/- mice or is the increased survival due to more active T cells? The gene expression data would lead one to predict that there would be an influx of CD4 and CD8 T cells into the marrow of the Sipa-/- mice, is this true?

Thank you for another very important comment. Based on the suggestion, we performed FACS analysis of the BM cells in Wt and Sipa1^{-/-} mice. We found that the numbers of CD4⁺ T cells and more prominently CD8⁺ T cells were significantly greater in the BM of Sipa1^{-/-} mice than that of Wt mice at day 12, when the rejection of CML cells became evident in Sipa1^{-/-} mice. The increase was mostly attributed to memory (CD62^{low} CD44^{high}) T cells, suggesting the enhanced T cell influx in the Sipa1^{-/-} host. The results are presented as

Fig. 4e. Since the expansion of CML cells occurred focally in BM, we further examined the topical association of T cells with the CML cells with immunostaining analysis. We confirmed that much greater numbers of T cells were associated with GFP⁺ CML cells in the Sip¹^{-/-} host than those in the Wt host. Also in the Sip¹^{-/-} BM, we often observed T cells forming a tight adhesion to the CML cells. The results are presented in **Supplementary Fig. 3**.

3. A major claim in the paper is that this immune mediated rejection of tumor in Sip¹^{-/-} mice is CML specific. This is based on the comparison of primary BCR-ABL transduced and BA-1 cells which are rejected to Wo-1 (T-ALL) and EL4 (Lymphoma). A major confounding factor in this interpretation is that both the primary transduced cells and BA-1 are made with an IRES-GFP while Wo-1 and EL4 lack GFP. It is highly likely that the rejection of these lines is due to recognition and targeting of GFP by the T cells. An experiment should be conducted to either express GFP in Wo-1 and EL4 OR remove GFP from the BCR-ABL construct and repeat the experiments.

This is a reasonable comment. However, please be noted that the Wo-1 cell line was derived from the thymic T-ALL developed from the BM pro-T cells transduced with C3G-F in the same IRES/EGFP retrovirus identical to that used in current experiments (Wang et al., Blood 2008). Thus, it also expressed GFP even more strongly than BA-1 cells as shown in the figure below. Therefore, we do not think that GFP plays a dominant role as a tumor antigen in the current system and so mention this in the revised text (p.17). Specific antigens involved in the current CML resistance in the Sip¹^{-/-} host remain to be verified. Neo-antigens derived from the Bcr-Abl fusion gene are the obvious candidates as reported in human CML patients (Ref. 18, 39, 40). It is also reported that self-antigen P3 proteinase, which is abundantly and exclusively expressed in normal myeloid cells can function as a potent CML antigen recognized by CD8⁺ T cells (Molldrem et al., Nature Med 2000). Notably, the P3-specific T cells are reported to develop only in certain immunotherapy such as allogeneic BMT and IFN α and show a high correlation to CML remission. Considering the proneness to systemic autoimmunity of Sip¹^{-/-} mice (Ishida et al., Immunity 2006), lineage-specific self-antigens such as P3 proteinase may be candidates. Verification of the specific antigens involved in the current system is within the scope of our next study.

Overall in summary the paper is very exciting and will add greatly to the field if the above issues are clarified.

Thank you for very thoughtful comments and valuable suggestions. I hope that we have addressed them adequately.

REVIEWERS' COMMENTS:

Reviewer #1 (Remarks to the Author):

I have reviewed the author's responses to the criticisms and have reviewed their revised manuscript, and I am satisfied that the authors were very responsive to the concerns.

Reviewer #2 (Remarks to the Author):

Review of "A potent immune mechanism eradicating chronic myelogenous leukemia-initiating cells uncovered by Sipa1 deficiency" by Xu et al. resubmission.

The authors have made significant changes and improvements to the manuscript. Thank you for adding the numbers of animals to each figure. It was not clear that the Wo-1 cell line also expressed GFP. Thank you for clarifying that point. Knowing that GFP is expressed by Wo-1 satisfies my concerns about foreign antigen expression. These changes improve the ability of the reader to interpret the manuscript. This report will add significant understanding to the field.

Responses to Reviewers' comments

Reviewer #1 (Remarks to the Author):

I have reviewed the author's responses to the criticisms and have reviewed their revised manuscript, and I am satisfied that the authors were very responsive to the concerns.

We thank the reviewer for very understanding comment and also appreciate the previous suggestions and comments, which very much helped us to improve our manuscript.

Reviewer #2 (Remarks to the Author):

The authors have made significant changes and improvements to the manuscript. Thank you for adding the numbers of animals to each figure. It was not clear that the Wo-1 cell line also expressed GFP. Thank you for clarifying that point. Knowing that GFP is expressed by Wo-1 satisfies my concerns about foreign antigen expression. These changes improve the ability of the reader to interpret the manuscript. This report will add significant understanding to the field.

We thank the reviewer for understanding and encouraging comment and also appreciate the previous suggestions and comments, which very much helped us to improve our manuscript.